# Precision dynamical mapping using topological data analysis reveals a hub-like transition state at rest

Manish Saggar [1] ✉, James M. Shine [2], Raphaël Liégeois[3,4], Nico U. F. Dosenbach [5] & Damien Fair [6]

In the absence of external stimuli, neural activity continuously evolves from one configuration to another. Whether these transitions or explorations follow some underlying arrangement or lack a predictable ordered plan remains to be determined. Here, using fMRI data from highly sampled individuals (~5 hours of resting-state data per individual), we aimed to reveal the rules that govern transitions in brain activity at rest. Our Topological Data Analysis based Mapper approach characterized a highly visited transition state of the brain that acts as a switch between different neural configurations to organize the spontaneous brain activity. Further, while the transition state was characterized by a uniform representation of canonical resting-state networks (RSNs), the periphery of the landscape was dominated by a subject-specific combination of RSNs. Altogether, we revealed rules or principles that organize spontaneous brain activity using a precision dynamics approach.

Spontaneous brain activity in the absence of sensory input is considered to be highly structured in both space and time[1] with amplitudes at least as large as stimulus-driven activity[2,3]. The ongoing patterns of cortical activity are thought to continually evolve over time and have been shown to encode multidimensional behavioral activity[4]. It is believed that the continuous evolution of cortical activity patterns could reflect multiple functions, namely, recapitulating (or expecting) sensory experiences[5–8], maintaining a rich repertoire of possible functional configurations[9,10], continuing top-down prediction/expectation signal for updating representation of the world[1], reflecting changes in the behavioral and cognitive states[11], and has been shown to be largely bistable[12–14]. However, it is not fully established whether transitions in intrinsic brain activity follow some underlying arrangement or instead lack a predictable ordered plan. Characterizing the rules underlying transitions in cortical activity has the potential to advance our understanding of the neural basis of cognition, and also to

better anchor psychiatric disorders onto more robust biological features[15–18].

Since its inception, functional magnetic resonance imaging (fMRI) has been used to non-invasively measure blood oxygen level-dependent (BOLD) signal as a proxy for neural activity[19]. Several fMRI studies have significantly advanced our understanding of brain functioning in healthy and patient populations by successfully identifying static or long-time-averaged measures of intrinsic functional organization[20–25]. To measure brain's intrinsic functional architecture, i.e., in the absence of any task (resting-state), co-fluctuations in the BOLD signal are assessed (a.k.a. resting-state functional connectivity). Although the dynamical aspect of brain activity has long been known to be critical in electrophysiology, low spatiotemporal resolution of the *human* neuroimaging has slowed down embracing dynamical analysis of the brain[26]. However, time-varying analysis of fMRI data is gathering momentum due to recent advances in data acquisition

[1]Department of Psychiatry and Behavioral Sciences, Stanford University, Stanford, CA, USA. [2]Brain and Mind Center, The University of Sydney, Sydney, NSW, Australia. [3]Institute of Bioengineering, École Polytechnique Fédérale de Lausanne, Lausanne, Switzerland. [4]Department of Radiology and Medical Informatics, Faculty of Medicine, University of Geneva, Geneva, Switzerland. [5]Departments of Neurology, Radiology, Pediatrics and Biomedical Engineering, Washington University School of Medicine, St. Louis, MO, USA. [6]Department of Pediatrics, University of Minnesota Medical School, Minneapolis, MN, USA. ✉e-mail: saggar@stanford.edu

methods, such as multi-band[27,28] and multi-echo[29] imaging that enhance spatiotemporal resolution of the acquired data and facilitate development of novel data analytics[30–38].

Time-varying analyses of intrinsic *human* neuroimaging data have revealed richer dynamics than previously appreciated, including the existence of fast switching between metastable states[39]; intermittent periods of globally coordinated co-fluctuations across spatially distributed brain regions[32,40,41]; large-scale metastable cortical waves[26,42]; and hierarchical temporal organization at the group level[36]. Further, individual differences in time-varying signals at rest have been associated with a wide range of cognitive and behavioral traits and even shown to be more sensitive than static (or averaged) functional connectivity[31]. Typically, a time-varying analysis first characterizes a set of brain states at the group level, followed by examining individual differences in frequency or duration of such states. A brain state is commonly defined as a transient pattern of whole-brain activation (or functional connectivity) and is usually characterized by activation of (or connectivity in) known large-scale brain networks (a.k.a. resting-state networks; or RSNs). Importantly, typical time-varying analyses (e.g., using sliding window-based approaches) have been prone to be affected by sampling variability and physiological artifacts in the fMRI data[43,44]. With that said, however, work using simultaneous wide-field optical imaging and whole-brain fMRI has established a direct link between resting-state hemodynamics in the awake and anesthetized brain and the underlying patterns of excitatory neural activity[45–47]. Thus, while the ongoing hemodynamics as measured by noninvasive fMRI are coupled to excitatory neural activity, novel methods are required to carefully parse neuronal dynamics while discounting artifactual transitions, with a goal towards deciphering the 'rules' that determine whole-brain transitions across brain states. For example, it is unclear whether the temporal transitions in brain activity (or connectivity) are best conceptualized as a continuous (or gradual) evolution[48–50] or discrete (or binary) switches[51–53]. Further, it is also unclear whether a transition from one so-called brain state to another is direct or does the brain pass through a set of intermediary states. Lastly, while previous work defined brain states at the group level, it is unclear whether individual differences exist in terms of the configuration of brain states.

The low spatiotemporal resolution and high complexity of the fMRI data make the study of whole-brain dynamics at the single person level ($n = 1$) a challenging endeavor. Specifically, the low signal-to-noise ratio of the BOLD signal[54] and the typically short duration of resting-state fMRI scans (~5–15 min[55]) impedes precise characterization at the individual subject level. Further, the high cost of MR data acquisition and excessive participant burden limit the amount of data that can be gathered. Fortunately, in the past few years, there is growing momentum towards collecting and sharing fMRI data using a precision functional mapping approach, where each participant is sampled at multiple occasions (>=10) yielding hours' worth of data for each individual[56–59]. Due to the vast heterogeneity in network topology from person to person, these approaches are critical to unveiling basic principles of brain function and organization. We argue that a similar approach for precision dynamics will be vital for deciphering the rules regarding how the *human* brain dynamically adapts from one configuration to the next and how these transitions relate to cognition and various psychopathologies[60–63].

In the current work, using a precision dynamics approach and the Midnight Scan Club (MSC) dataset[57], we aimed at revealing the overall landscape of at-rest whole-brain configurations (or states) at the single individual level. We hypothesized that by revealing and characterizing the overall landscape we could interpret the rules that govern transitions in brain activity at rest. The MSC dataset includes individually defined parcellations and ~5 h of resting-state fMRI data for each participant—both of which allowed us to examine the topology and dynamics of at-rest whole-brain configurations in unprecedented

detail. We also addressed previous methodological limitations by using tools from the field of topological data analysis (TDA), which are designed to learn the underlying topology (or shape) of high dimensional datasets that are relatively sparse and noisy[64,65]. Specifically, here, we used the TDA-based Mapper approach that generates the shape of the underlying dataset as a graph (a.k.a. shape graph)[34,66,67]. Mapper has been previously shown to capture task-evoked transitions in the whole-brain activity patterns at the highest available spatiotemporal resolution, limited only by acquisition parameters[33]. Unlike previous time-varying analytics, Mapper does not require splitting or averaging data across space or time (e.g., windows) at the outset. Further, Mapper does not require any a priori knowledge about the number of whole-brain configurations and does not impose strict assumptions about mutual exclusivity of brain states[39]. Lastly, the presented results were not only validated in the MSC dataset using split-half analysis but were also independently validated using a separate dataset from the Human Connectome Project[27] ($n = 100$, unrelated individuals).

## Results

### Estimating reliable landscape of whole-brain configurations at the single participant level

Our first aim was to utilize the TDA-based Mapper approach to reliably estimate individually specific landscape (or manifold) of whole-brain configurations. To ensure the replicability of our findings, we first split the MSC data for each participant into two halves (discovery and replication sets)—each with ~2.5 h of data per participant. Thus, for each participant, out of a total of ten sessions (each 30 mins long), we assigned odd sessions to the discovery and even sessions to the replication set.

After rigorous preprocessing (see Methods and Gordon et al.[57] for details), the individually specific parcellated data were fed into the TDA-based Mapper pipeline[33], which consists of four main steps. First, the high-dimensional neuroimaging data are embedded into a lower dimension d, using a non-linear filter function f. Importantly, information loss incurred during dimensionality reduction is putatively recovered during the partial clustering step[34,66] (the third step in the Mapper pipeline). To better capture the intrinsic geometry of the data, a nonlinear filter function based on neighborhood embedding was used[33] (see Methods for benefits of this non-linear approach). Second, overlapping d-dimensional binning is performed to allow for compression and to putatively increase reliability (by reducing noise-related perturbations). Third, partial clustering within each bin is performed, where the original high dimensional information is used for coalescing (or separating) data points into nodes in the low-dimensional space and hence allows for partially recovering information loss incurred due to dimensionality reduction. Lastly, to generate a graphical representation of the data landscape, nodes from different bins are connected if any data points are shared between them. Fig. S1 provides a step-by-step representation of the Mapper pipeline.

In contrast to traditional graphical representations of neuroimaging data, nodes in the Mapper-generated shape graph represent clusters of highly similar whole-brain volumes (or time frames (TRs)), and edges connect any two nodes that share one or more whole-brain volumes. This approach naturally embeds temporal patterns within the spatial structure of the graph, which in turn confers several benefits for interrogating the spatiotemporal characteristics of the resting brain. For instance, using this shape graph, we can track how the resting brain dynamically evolves across different functional configurations at the individual-subject level. Importantly, our approach does not require any time-window averaging, which could potentially blur the data and has been shown to lead to artifactual findings due to head movement artifacts and sampling variability[43,44].

To reveal the rules that govern transitions between whole-brain configurations at-rest, we examined: (a) the topological properties of

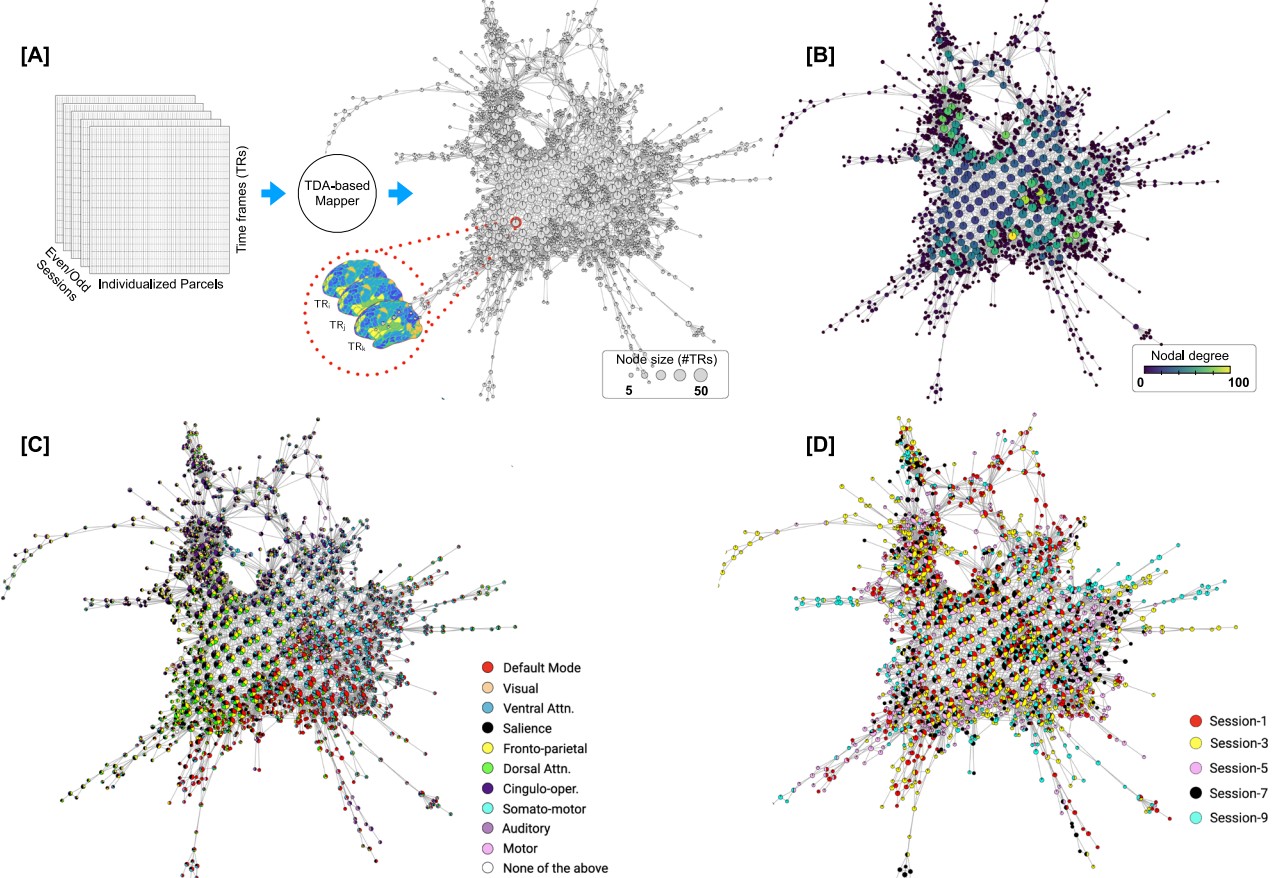

**Fig. 1 | Estimation and characterization of the dynamical structure underlying transitions in intrinsic brain activity using our TDA-based Mapper approach.** Here, we present data from a representative participant (MSC-01; odd sessions). **A** Individualized parcellated data from the highly sampled Midnight Scan Club (MSC) individuals[57] was split into two halves: odd sessions (2.5 h) and even session (2.5 h) sets. The Mapper approach was independently run on each set to generate the underlying structure as a graph. Each graph consists of nodes and edges, where the nodes could in turn contain multiple whole-brain volumes (or TRs; the size of a node represents the number of TRs). The nodes are connected if they share TRs. **B** The Mapper-generated graph can be characterized in several ways. Here, we examine topological properties by annotating the graph nodes using nodal degree. **C** The graph can also be annotated with meta-information to characterize the mesoscale structure. Here, we show annotation using the activation of individual-specific resting-state networks (RSNs). A pie-chart-based annotation is used to reveal the proportion of time frames with each node belonging to different RSNs. **D** Similarly the graph can also be annotated using other available meta-information, e.g., session information.

the shape graph, such as the degree distribution and existence of hubs; (b) the relationship between the Mapper embedding and canonical resting-state networks; and (c) the transitions between whole-brain configurations. See Fig. 1 for our analytical approach. In addition to individual variability in the characteristics of Mapper-generated landscapes, we also report the central tendency (or group average) of the dynamical landscape at rest. To account for linear properties of the data (e.g., serial auto-correlation) and sampling variability issues, we compared results with two null models, namely, the phase randomized null[68] and the multivariate autoregressive null model[44]. Lastly, the results revealed from the MSC dataset were independently validated using a separate dataset from the Human Connectome Project[27] (HCP; $n = 100$ unrelated individuals).

## Topological properties of the landscape reveal the existence of hubs

We first characterized the Mapper-generated graphs by calculating nodal degree, which measures the strength (or number) of connections (or edges) per node. In the context of the shape graph, high degree nodes represent whole-brain activation patterns that are shared by many other nodes (i.e., are visited often in the temporal evolution of the data). The degree distribution for each participant and their corresponding splits (odd and even sessions) were further

characterized to determine whether it deviated in any way relative to what might be expected by linear properties of the data (e.g., auto-correlation in the BOLD signal). We accomplished this goal by comparing the degree distribution from the real data with multiple instances of the two pre-defined null models (phase randomization and multivariate AR model). As evident from the degree distribution plots (Fig. 2A), the real data contained heavy (or fat) tail distributions as compared to both null models. The heavy tail distribution is iconic for most real-world networks and indicates the existence of highly connected nodes[69–72]. This finding was independently replicated in both halves of the MSC data. Statistical difference in the proportion of high-degree nodes (>20) in the real versus null data was assessed using one-way ANOVAs for both odd ($F(2,27) = 6.27$, $p = 0.0058$) and even sessions ($F(2,27) = 14.49$, $p = 5.32 \times 10^{-05}$).

Highly connected nodes that are also topologically central (i.e., influential) in the graph are known as hubs. Hubs are hypothesized to act as focal points for the convergence and divergence of information in the network[72]. The existence of hubs in the Mapper-generated graph would indicate the presence of nodes (or whole-brain configurations) that are visited often, potentially as intermediate (or transition) states. To examine the existence of hubs in the Mapper-generated landscapes, we estimated the closeness centrality of highly connected nodes[73,74]. This measure associates

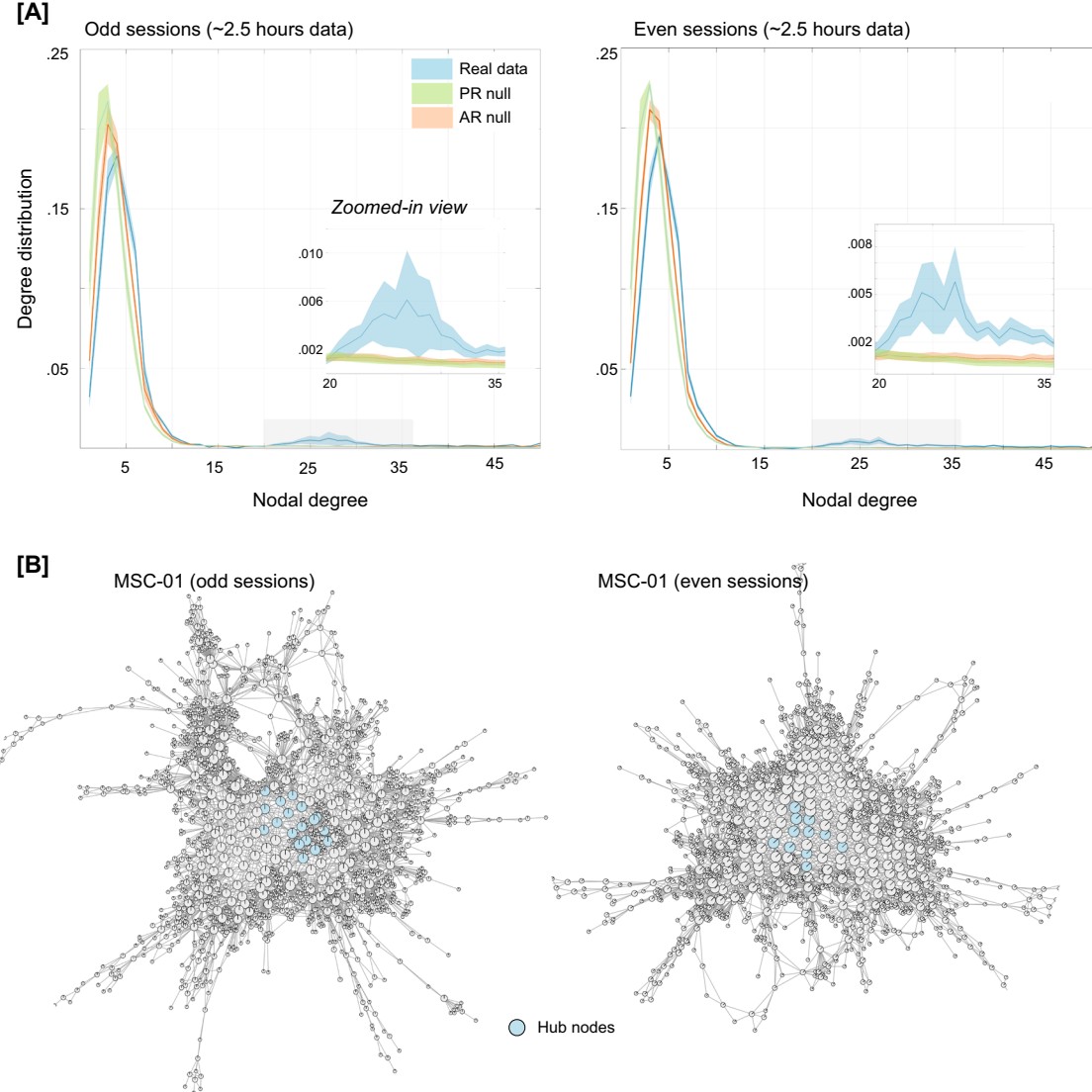

**Fig. 2 | Characterizing the Mapper-generated graph using degree distribution.**
**A** Degree distributions averaged across the ten participants, separately for odd and even sessions. For examining linear vs. nonlinear aspects, two null models were used, namely, the phase randomized null and the multivariate autoregressive null model. As evident from the degree distribution plots, real data show a significantly fat tail (>20) as compared to both nulls. This finding was independently replicated in both halves of the data. The shaded area represents standard error around the mean (SEM). **B** Show Mapper-generated graphs for a representative participant (MSC-01), highlighting nodes that act as hubs (i.e., nodes with high degree (>20) and high centrality (top 1%)). Similar plots were observed across all MSC dataset participants (see Fig. S2).

the nodes with the shortest average path lengths as being the most influential (or central) for the graph. Nodes with high closeness centrality can receive information from other parts of the network in a short time (and vice versa). Across both halves of the data and all participants, the topologically central highly connected hubs were found to occur in the shape graph (Fig. 2B highlights the hubs in a representative participant, and supplementary figure Fig. S2 shows hubs across all MSC participants).

Although data censuring was done to reduce the impact of head movement related artifacts, we additionally examined whether the presence of high degree nodes (and hubs) was associated with head movement or global signal variations. No difference in framewise displacement (FD) or global signal was observed between brain volumes represented by high and low degree nodes of the shape graph (ps > 0.05 for FD and global signal), for either split of the data. Further, parameter perturbation analysis was performed to make sure topological properties of the graph were stable across a moderate range of Mapper parameters (see Methods and supplementary Fig. S6). Similar

work was previously done to show Mapper-generated graphs were stable across different parameter combinations[33].

The presence of hubs in the dynamical landscape (across all participants) provides evidence for whole-brain configurations that (i) are often visited during rest; (ii) are highly conserved at the individual subject level; and (iii) may act as a 'switch' between different configurations to putatively organize the spontaneous activity during rest.

### Hubs represent uniform (mean) activation across all RSNs, whereas peripheral nodes represent increased activation in one (or more) RSNs

To relate Mapper-generated graphs to canonical neuroanatomical depictions of the resting brain, we annotated nodes in the Mapper graph using the relative engagement of a set of canonical large-scale resting-state networks (RSNs). Importantly, we leveraged a set of individually defined network assignments that were pre-calculated for individuals in the MSC dataset[57]. Figure 3A, B shows a Mapper-generated graph for a representative participant (MSC-01), where

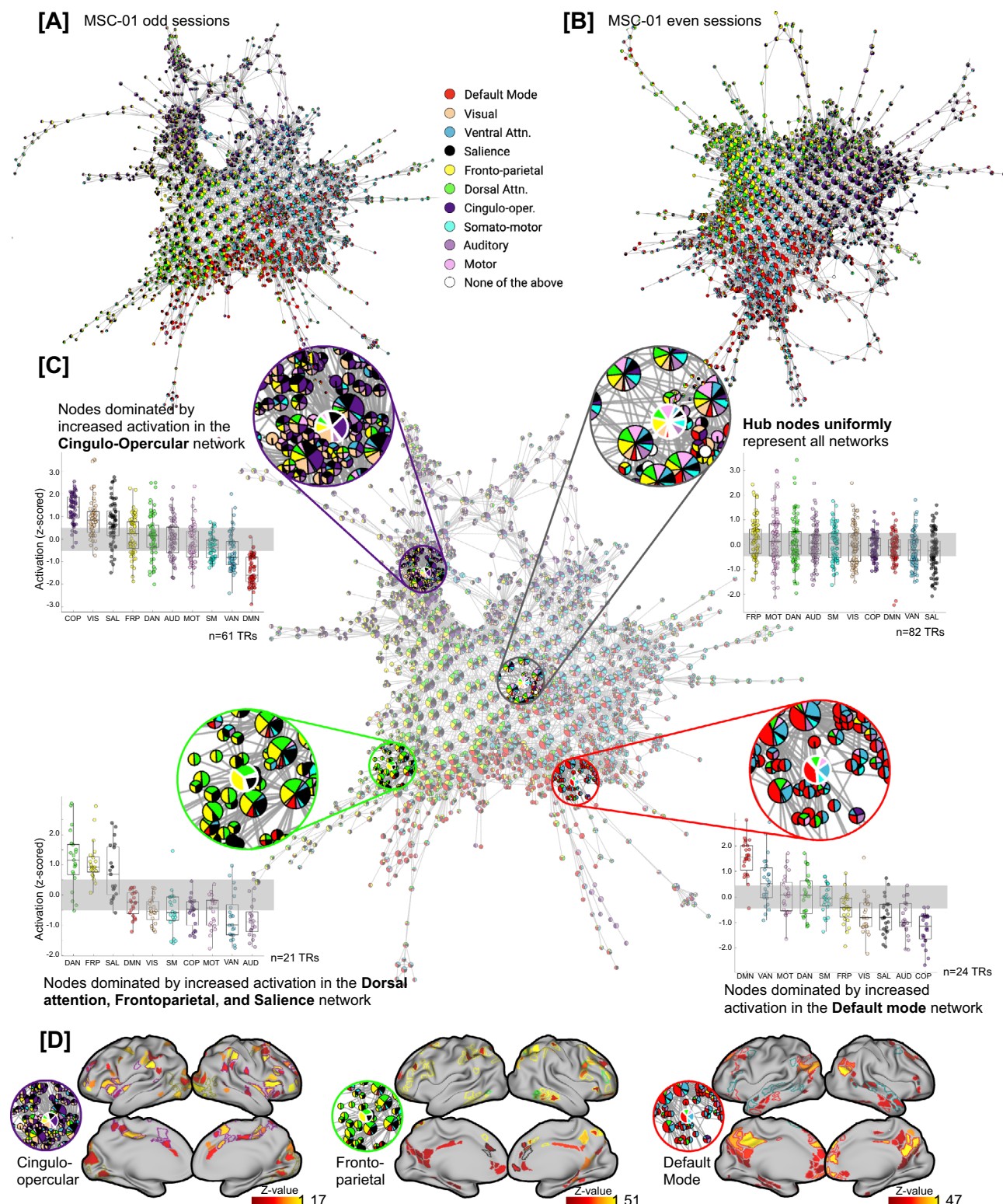

each node is annotated by activation in the RSN. In this view, each node is annotated using a pie-chart notation to show the proportion of brain volumes (or TRs) that have any RSN activated (above a certain threshold). The mean signal for each RSN was z-scored and a threshold of 0.5 SD above the mean was used to denote activation of an RSN (other thresholds produced similar results).

As shown in Fig. 3C, the topography of the Mapper-generated landscape provides important insights into the temporal architecture of the resting brain. Topologically highly connected and

central hubs contained brain volumes in which no characteristic RSN was activated above the mean, whereas nodes with brain volumes dominated by one (or more) RSN(s) tend to occupy the peripheral corners of the landscape. The maps for all individual subjects demonstrated this same basic pattern, although there was evidence to suggest that different combinations of RSNs were dominant in different individuals. For instance, the default mode, frontoparietal, and cingulo-opercular clearly dominated the periphery of MSC-01 landscape, across both splits of the data, but other

**Fig. 3 | Annotating Mapper-generated graphs based on individual-specific large-scale resting-state networks (RSNs). A, B** Shows Mapper-generated graph for a representative participant (MSC-01; **A** odd and **B** even sessions). Here, each node is annotated by activation in the known large-scale resting-state networks. Each node is annotated using a pie-chart to show the proportion of RSNs activated within each node. As evident, for MSC-01, for both odd and even sessions, the Mapper-generated graph has mainly three networks dominating on the periphery of the dynamical landscape: default mode, frontoparietal, and cingulo-opercular networks. **C** Zoomed-in view of the Mapper graph generated using MSC-01 odd sessions. The nodes with dominating RSNs are located more towards the periphery of the landscape, while the hubs of the landscape are not dominated by any RSN

and rather have uniform mean-level distribution across all RSNs. Four zoomed-in circles highlight four exemplary nodes, where the peripheral nodes have one (or more) RSNs in the majority and the central node has no network dominating. Box plots represent activation (z-scored) in the corresponding RSNs across all time frames (TRs) within each highlighted node. In each boxplot, the box denotes interquartile range (IQR), the horizontal bar indicating the median, and the whiskers include points that are within $1.5 \times$ IQR of upper and lower bounds of the IQR (25th and 75th percentiles). **D** Presents mean whole-brain activation maps for each of the three peripheral nodes thresholded at $z = 0.5$. Borders for individual specific RSNs are highlighted. As evident, whole-brain activation maps of each peripheral node clearly show higher activation in the corresponding RSNs.

participants had a different combination of networks dominating their landscapes (Fig. S3).

## RSN-based topography of landscapes is highly subject-specific and stable across sessions

To quantify the subject-specificity and examine whether Mapper-generated landscapes were stable within participants, we computed similarity between RSNs in terms of their co-localization on the Mapper-generated graphs. If two networks are co-localized on the graph, then they activate (or deactivate) synchronously. Figure 4A, B presents network similarity matrices for three representative participants across their odd and even sessions. As evident, qualitatively, the network similarity matrices are comparable across odd and even sessions. To quantify subject specificity in terms of network similarity, we compared network similarity matrices across sessions and participants using Pearson's correlation. As evident in Fig. 4C, high within-participant correspondence (i.e., the high similarity between odd and even sessions) for network similarity matrices was observed as compared to between participant correspondence ($F(1,198) = 39.36$, $p = 2.18 \times 10^{-09}$), suggesting dynamical landscapes are subject-specific and stable over sessions.

Lastly, we computed the central tendency of the dynamical landscape topography by averaging the network similarity plots across participants. As evident in Fig. 4D, the group averaged topography could present a putatively different picture than the individual topographies. Across both halves of the data, group-averaged topography represents less synchrony between higher-order cognitive networks (e.g., default mode, fronto-parietal, etc.) than unimodal sensorimotor networks (e.g., visual, auditory, etc.). However, this discrimination between network types could be due to group averaging and is not necessarily present at the individual participant level. At the participant level, subject-specific combinations of higher-order cognitive networks and unimodal sensorimotor networks are observed to be in synchrony. In summary, individual subjects demonstrated idiosyncratic, yet highly replicable, topological signatures at the level of canonical resting-state networks.

In addition to RSN-based topography, subject specificity was also observed in terms of the topological properties of the Mapper-generated landscapes. For example, the degree distribution of Mapper-generated graphs was more similar between sessions within a participant than across participants ($F(1,18) = 5.31$, $p = 0.034$). Also, the proportion of hubs was similar across splits of the data (i.e., odd vs. even sessions; $F(1,18) = 1.73$, $p = 0.2$). Thus, suggesting, both RSN-based topographical and traditional topological properties of the Mapper-generated landscapes were subject-specific and stable across sessions.

## Traversal on the Mapper-generated landscape revealed a topographic gradient with hubs representing a putative transition state

Next, we used a variance-based approach to examine whether the traversal on the landscape—i.e., going from one corner to the next (or towards the center)—was smooth (i.e., continuous) or bumpy (i.e., discrete). To this end, we estimated the mean activation for each RSN

(across all the brain volumes) within each node, followed by estimating variation (standard deviation; S.D.) in the mean network-level activation across all RSNs. High variance (or S.D.) indicated the dominance of one or more RSNs, whereas low variance indicated uniformity across mean RSN activation. As shown in Fig. 5A (using a representative participant, MSC-01), annotating Mapper-generated graphs using this variance-based approach revealed a topographic gradient in the dynamical landscape (Fig. 5B), where the peripheral nodes had higher variance with a continual decrease in variance when going towards the center of the graph.

To further illustrate the gradient between peripheral dominating nodes and central hub (non-dominating) nodes, using MSC-01, Fig. 5A shows three trajectories (one for each of the three dominating networks) and the corresponding boxplots for a sample of nodes from each trajectory – starting from the dominating node on the periphery and moving towards the hub (or non-dominating) nodes. We also present mean cortical activation for several nodes in the three trajectories. As evident, peripheral nodes represent time frames where one or more RSN was more activated than others, while as one traverses towards the center of the graph the nodes represent time frames with uniform mean-level activation across all RSNs. Figure 5C shows average distribution of S.D. values, over ten MSC participants, for hubs (blue) and other nodes (orange). As evident, the hubs had significantly lower S.D. values than non-hub nodes (for both splits of the data; odd: $F(1,18) = 141.84$, $p = 5.70 \times 10^{-10}$ and even: $F(1,18) = 222.20$, $p = 1.49 \times 10^{-11}$)—suggesting uniform distribution across all RSNs. Similar gradients were observed across all ten MSC participants (Fig. 5B and Fig. S4).

To confirm whether the brain configuration represented by the hubs does indeed act as a putative switch, we examined changes in brain activation patterns in the time domain, i.e., at the single time frame (or brain volume) level. The RSN-based proportions from each graph node were propagated to the individual time frames (or TRs) represented by that node. For nodes dominated by any particular RSN, the encompassing TRs were assigned the dominant RSN. For hubs, where RSNs were uniformly distributed, the encompassing TRs were assigned a new label (hub state). Figure 6A depicts labels for each TR, across the ten MSC participants, separately for the two splits of the data. To better characterize transitions in RSN-based states we estimated the discrete-time finite-state Markov chains[75] for each participant and data half. Note the strong visual similarity between rows of the two session matrices. Figure 6B shows transition probabilities estimated from the Markov chain estimation averaged across all participants, separately for the two splits of the data. While estimating Markov chains and associated transition probabilities, we ignored putatively artifactual transitions associated with frames discarded due to head movement and due to stitching the sessions together. As evident from the estimated transition probabilities, brain configuration represented by the hubs (or our putative transition state) was observed to be the most sought-after destination from any other RSN-dominated state. Figure 6C shows the same result at the individual participant level, such that from any other RSN-dominant state the brain was more

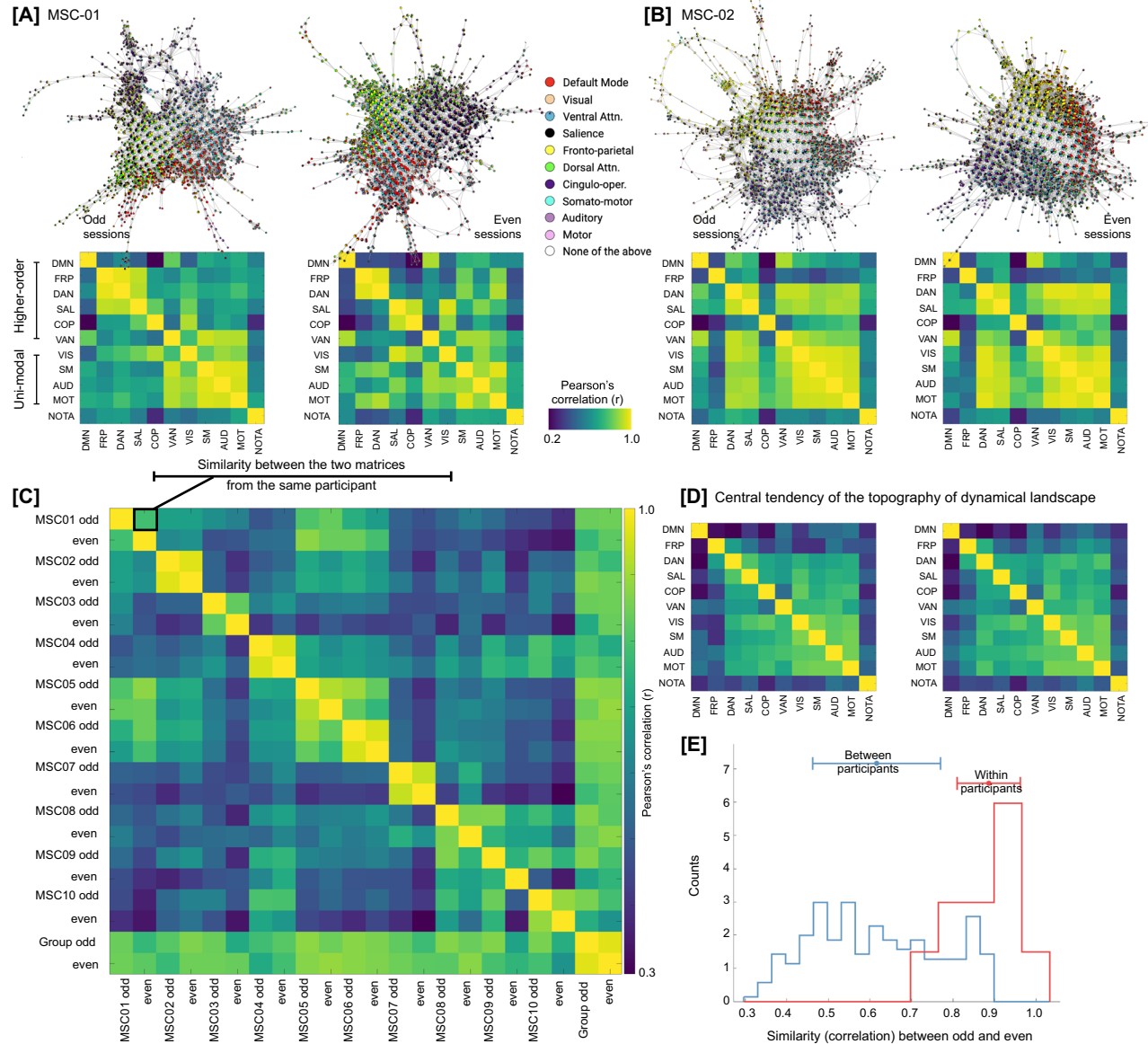

**Fig. 4 | Dynamical landscapes and their RSN-based topography are subject-specific. A, B** Shows Mapper-generated graphs annotated by RSN activation for two representative participants (MSC01-02). Both split halves (odd and even sessions) are shown for each participant. For each half, the figure also shows a similarity (correlation) matrix between RSNs, where a high correlation between two RSNs suggests co-location (or co-activation) on the Mapper-generated graph. As evident through Mapper-graph annotations and between network correlations, there was a high degree of similarity between two halves of the same participant. **C** To quantify between- vs within-participant correspondence across network similarity matrices, network similarity matrices were compared across split halves from all participants. As shown in the between-subject matrix, high correspondence was observed for within-participant matrices, suggesting dynamical landscapes demonstrated idiosyncratic, yet highly replicable, topological signature at the level of canonical resting-state networks. **D** Central tendency of the dynamical landscape, averaged over ten highly sampled individuals, for odd and even sessions. **E** RSN-based topography was highly similar within participants, as compared to between participants (one-way ANOVA: F(1,198) = 39.36, $p = 2.18 \times 10^{-09}$).

likely to transition to the hub transition state – providing evidence for the hub state to be a likely intermediary between any two RSN-dominating states. Transition probabilities can also be represented as a graph (shown in Fig. 6D). Lastly, we observed the transition probabilities to be highly subject-specific and reliable across sessions (Fig. 6E). A one-way ANOVA showed transition probability matrices across the two halves of data were more similar within participant (highly correlated) than across participants (F(1,398) = 63, $p = 2.13 \times 10^{-14}$).

To further confirm the transitional and continuous interplay between hub states and RSN-dominated states, we examined whether the hub states appear at the tail ends of RSN-dominance in the time domain (i.e., at the level of individual brain volumes). For this

analysis, instead of propagating the RSN-dominance vs. hub state dichotomously into the time domain (i.e., labeling every TR with dominating network or a hub state), we propagated mean activation values of dominating RSN Mapper nodes to the timeframes. The continuous evolution of RSN dominance was observed at the timeframe level and hub states were found more likely to be present at the tails of RSN dominance—providing further evidence for the transitory nature of hub states (Fig. S7A–C). To quantify this inverse relation between RSN dominance and hub states, we estimated temporal correlation between RSN mean amplitude and hub state occurrences across participants. Predominantly negative relations were observed between the two for all participants and across sessions (Fig. S7D), suggesting that the hub states tend to appear in

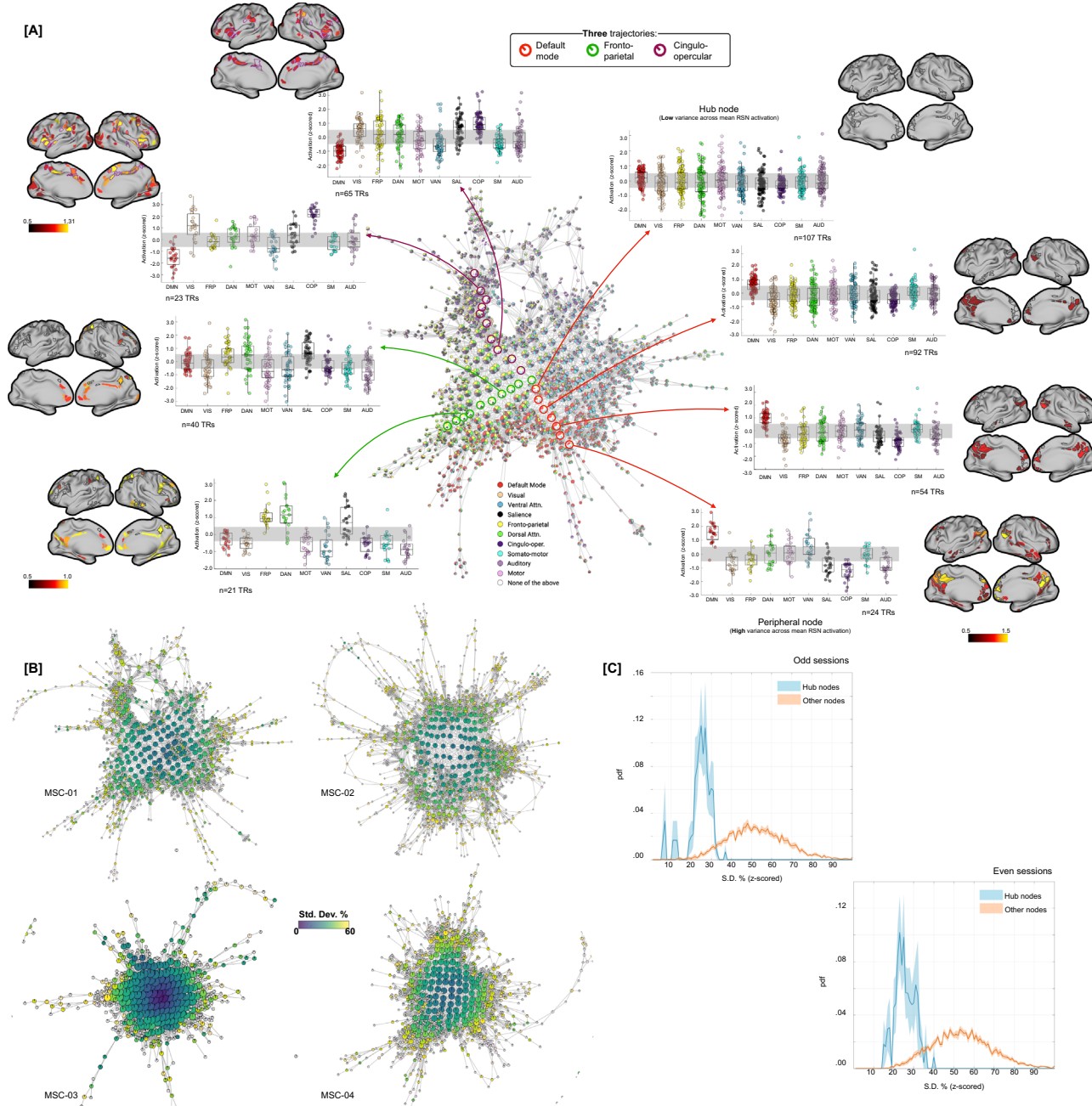

**Fig. 5 | Annotating the traversal on Mapper-generated landscape using a variance-based approach revealed a dynamical *topographic gradient*.**
**A** Depicting traversal on the Mapper-generated landscape from peripheral (RSN-dominated) nodes towards centrally located hubs. Three putative trajectories are highlighted on the Mapper graph, corresponding to three dominating RSNs for MSC-01 participant. For all three trajectories, activation across RSNs (as box plots) and mean whole-brain activity (on cortical surfaces) is shown for multiple nodes. As evident, peripheral nodes are dominated by activation in one of the RSNs and traversal towards the hubs result in reduced RSN activity. In each boxplot, the box denotes interquartile range (IQR), the horizontal bar indicating the median, and the whiskers include points that are within 1.5 × IQR of upper and lower bounds of the IQR (25th and 75th percentiles). **B** Annotating Mapper-generated graphs using

variance-based approach, i.e., coloring nodes based on the amount of variance (or SD) across mean RSNs activation, revealed a dynamical *topographic* gradient. Here, we show variance-based annotation of Mapper graphs for four participants from the MSC dataset (odd sessions). The topographic gradient was observed consistently across participants and for both even and odd sessions (see Fig. S4). **C** Group averaged distribution of SD values, over ten MSC participants, for hubs (blue) and other nodes (orange) is shown, with SEM as shaded value. Evidently, the hubs had significantly low variance across mean RSN activation (indicating uniformly distributed RSN), while the non-hub nodes were highly variant across mean RSN activation. The brain overlays were created by the authors using Connectome Workbench Software (https://www.humanconnectome.org/software/connectome-workbench).

the tails of RSN dominance and putatively trigger transitions between RSNs.

In summary, traversal directly on the Mapper-generated landscape revealed a continuous evolution of brain dynamics—a dynamic topographic gradient. Similar traversal in the time domain (at single

frame level) revealed that the brain configurations represented by hubs acted as a putative switch (or a transition state) between different RSN-dominated configurations. Further, the transition probabilities between states were individual-specific, indicating a putative future application in precision medicine.

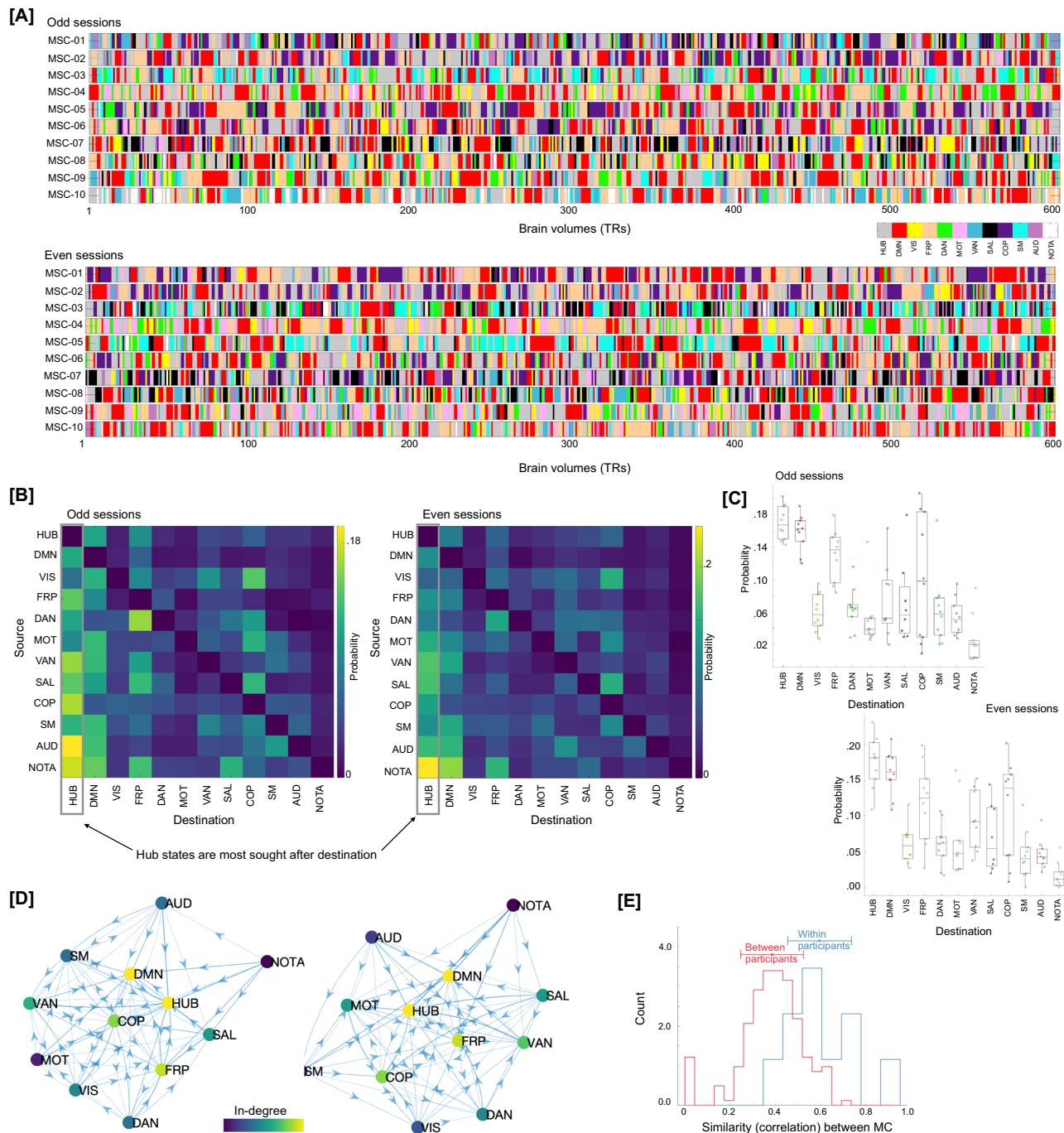

**Fig. 6 | Traversal in the temporal domain at the single frame level. A** Depicts transitions in brain activation over time frames in terms of dominant individual-specific RSN (or hub-like state). Each time frame (or TR) was labeled from the Mapper-generated shape-graphs by propagating the RSN-based annotation from each graph node to the time frames represented by that node. In addition to RSNs, a new label representing hub-nodes was also generated. As evident, hub state was often visited by participants across both data splits. Only showing a subset of timeframes (first 600 frames) for each participant for ease of viewing. **B** A discrete-time Markov chain was estimated using RSN-based labels for each participant and data split. While estimating transition probabilities, transitions due to discarding of motion affected frames and stitching sessions were rejected. Here, we present transition probability matrix averaged over all 10 MSC participants. Diagonals were suppressed to better illustrate transition probabilities across states. The hub state was observed to be the most sought-after destination from any other state, followed by the default mode network. **C** Boxplots depicting high probability of transitioning into the hub state from any other state, across all participants (*n* = 10 individuals). In each boxplot, the box denotes interquartile range (IQR), the horizontal bar indicating the median, and the whiskers include points that are within 1.5 × IQR of upper and lower bounds of the IQR (25th and 75th percentiles). **D** Estimated Markov chain averaged across all participants. As evident, the hub-state was observed to be most central and with highest in-degree. **E** The transition probability matrices (as show in **B**) were reliably estimated at the individual participant level (i.e., high within-participant similarity), indicating a putative application in precision medicine approaches.

### Replicating main results in an independent dataset

Although split-half data validation was performed for the MSC dataset, we further replicated the main results in an independent multi-session resting state fMRI dataset (100 unrelated participants from the Human Connectome Project (HCP)[27]. In the HCP dataset, four 15 min sessions of resting state scans were acquired over a period of two days. Thus, for each individual, we could analyze up to 1 h of resting state fMRI data. Important to note that the HCP data was substantially lower in scan duration than the MSC dataset (with 5 hours of resting state fMRI data per individual). Further, instead of using individually-defined parcellation, we used a group parcellation (Gordon atlas with 333 brain regions[76]).

After generating Mapper landscapes for each HCP participant, we first compared the degree distribution of graphs generated from real versus null data (from phase randomization and multivariate AR models). Like the MSC data, the HCP data also showed heavy (or fat) tail distributions as compared to both null models. Statistical difference in the proportion of high-degree (>20) nodes in the real versus null data was assessed using one-way ANOVA (F(2, 225) = 288.11, $p = 8.88 \times 10^{-63}$; Fig. 7A). Mapper-generated landscapes from the HCP data also contained hub-nodes (Fig. 7B).

Next, we annotated Mapper-generated graphs using the relative engagement of a set of canonical large-scale resting state networks (RSNs). As opposed to individually-defined networks for the MSC dataset, we used a group parcellation (Gordon atlas with 333 brain regions[76]) for the HCP data. Results are shown for three representative participants in the Fig. 7C. We observed highly connected and central hubs contained brain volumes where no particular RSN was activated, whereas nodes with brain volumes dominating from one particular RSN tend to occupy the peripheral corners of the landscape. The maps for individual subjects all demonstrated this same basic pattern, although there was evidence to suggest that different combinations of RSNs were dominant in different individuals.

Lastly, for the HCP dataset, we examined traversal on the landscape as well as temporal evolution of brain activation patterns at the single time-frame level. Using a variance-based approach, like the MSC-dataset, we again observed a smooth topographic gradient in the dynamical landscape of HCP participants, where the peripheral nodes had higher variance with a continual decrease in variance when going towards the center of the graph (Fig. 7C, D). For the temporal evolution of brain activation patterns at the single TR level RSN-based proportions from each graph node were propagated to the individual time frames (or TRs) represented by that node. Figure 7E depicts RSN-based labels for each TR, across the 30 representative HCP participants (randomly first 30 were chosen). Using discrete-time finite-state Markov chains, we also estimated transition probabilities, while ignoring putatively artifactual transitions associated with frames discarded due to head movement and due to stitching together sessions. In parallel to the MSC data, the HCP data also provided evidence for the hub-state to be the most sought-after destination from any other RSN-dominated state; thereby providing a putative role of intermediating between other RSN-dominating states (Fig. 7F,G).

## Discussion

Understanding how the brain dynamically adapts its distributed activity in the absence of any extrinsic stimuli lies at the core of understanding cognition. Although several innovative approaches have been developed to study the dynamical properties of intrinsic (or at rest) brain activity, the organization principles governing transitions in spontaneous activity are not fully understood. For example, it is unclear whether transition from one brain state to another is direct, or whether the brain passes through a set of characteristic intermediary states. Further, while previous work defined brain states at the group level, it is unclear whether individual differences exist in terms of how the brain states themselves are configured. Lastly, more work is

needed to understand whether temporal transitions in brain activity are best conceptualized as a continuous or discrete. To address these foundational questions, using a precision dynamics approach at the single participant level, we constructed the overall landscape of the whole-brain configurations at rest. Altogether, four robust findings were observed: (1) across all participants, the landscape of the whole-brain configurations contained centrally located hub-nodes that were often visited and likely acted as a switch or transition state between different configurations to organize the spontaneous brain activity; (2) transitions occurred as a smooth dynamic topographic gradient in the landscape, suggesting a continuous (as opposed to discrete) setup for brain state transitions at rest; (3) importantly transition probabilities between one state to another, at the level of a single time frame, were subject-specific and provided a stable signature of that individual; and (4) while the hub-nodes were characterized by a uniform representation of canonical RSNs, the periphery of the landscape was dominated by a subject-specific combination of RSNs (which was also stable across sessions). All the findings reported in this work were corroborated using a split-half validation and replication in an independent dataset. Together, using precision dynamics approach we revealed several rules or principles organizing spontaneous brain activity.

We begin the discussion by first providing a coarse viewpoint of our results that aligns well with previous and more recent works that have identified brain dynamics at rest as a bistable phenomenon. We then dive deeper into the rich subject-specific idiosyncrasies that our work revealed as our approach allowed precision analytics. We then provide a discussion on how our approach can putatively address common limitations of the previous work. Lastly, we provide limitations of our work and avenues for future applications.

From a coarse vantage point, the presence of low-amplitude (or close to mean activation) hub configurations versus high-amplitude peripheral configurations points towards bistable brain dynamics at rest. This bistable phenomenon is in line with the previous theoretical[12–14] and recent empirical work that has also shown brain dynamics during resting state to be predominantly bistable[36,77,78]. In contrast to the null models, real data showed existence of significantly higher number of hubs that were centrally located in the landscape and were representing whole-brain configurations with mean-level activity across all RSNs. The periphery of the landscape, on the other hand, was representative of one (or more) dominant RSNs.

Using Hidden Markov Models (HMM), van der Meer and colleagues recently reported brain dynamics during rest to be primarily driven by whole-brain configurations where all RSNs were uniformly expressed with amplitude close to mean network activities, while configurations with one (or more) dominant RSNs were only evident sporadically[77]. At the coarse level, our results are in line with these findings as we also observed intrinsic brain activity to be largely driven by whole-brain configurations with uniform RSN representation (i.e., hub-nodes), while configurations with one (or more) dominant RSN (i.e., peripheral nodes) evident sporadically. However, it is important to note that we used precision connectomics data (with longer duration scans) and individual-level definition of brain configurations (as opposed to group-level in case of HMM). These data and methodological enhancements led us to examine finer details about resting brain dynamics as detailed in the next sub-section.

In another work, also using HMMs, Vidaurre and colleagues found that transitions in intrinsic brain activity is stochastic and cycles between two major meta-states, where the first meta-state was associated with unimodal networks (i.e., sensorimotor) and the second meta-state involves regions related to higher order cognition[36,79]. Across individuals, the authors observed one of the two meta-states to be dominating, such that the brain cycled between networks within a meta-state more frequently than networks across meta-states. To anchor the topographical properties of the observed landscape of whole-brain configurations, we computed similarity between RSNs in

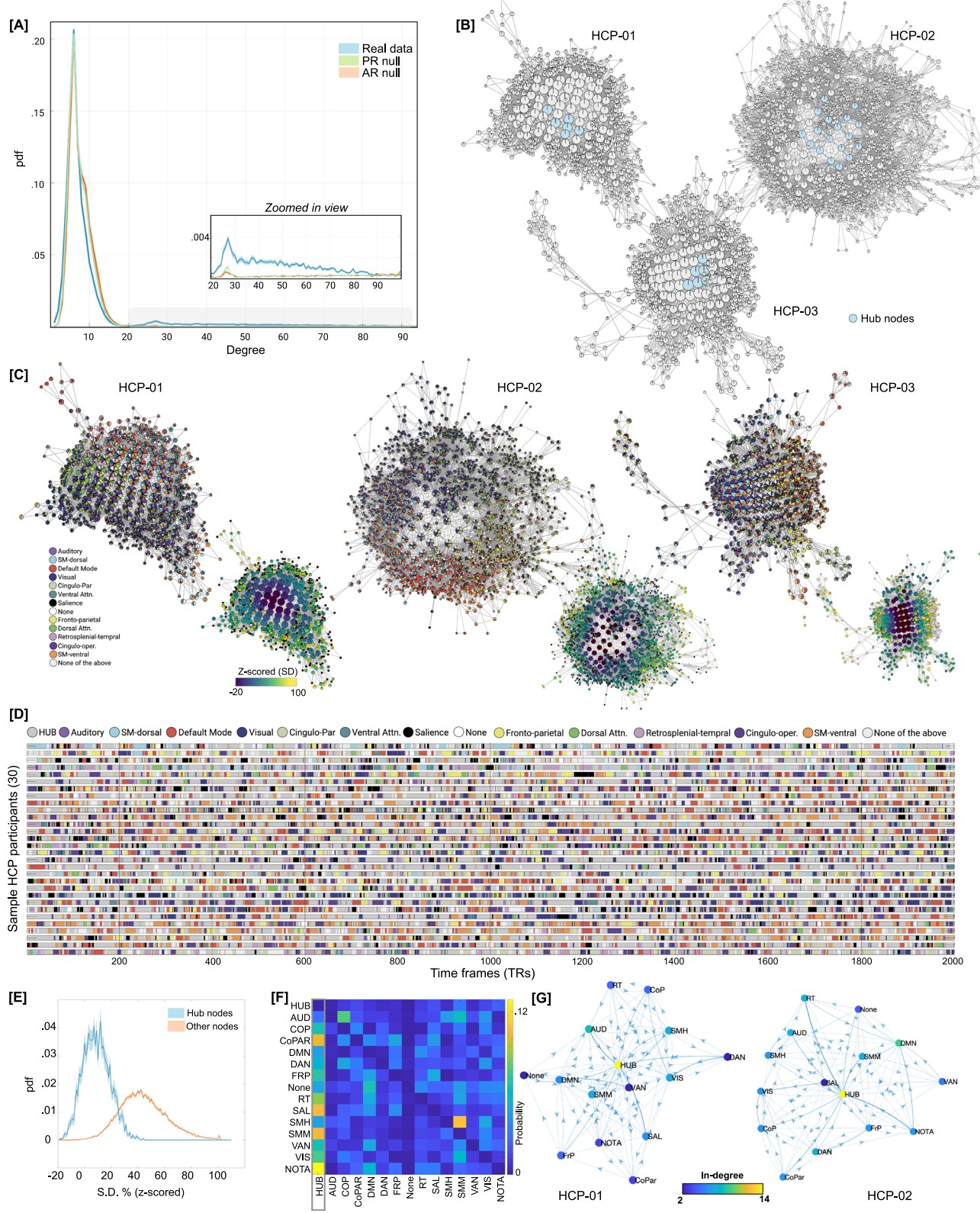

terms of their co-localization on the Mapper-generated graph. Co-localization of two networks on the Mapper-generated graph implies higher chances of co-activation (or co-fluctuation). As shown in Fig. 4D, at the group-level, we also observed a hierarchy of network co-locali-zation, broadly separating unimodal sensorimotor and higher-order cognitive networks. This group-level hierarchy was stable across ses-sions. However, we also observed individual differences in network co-

localization that were highly subject-specific and not exactly following the hierarchy between unimodal and higher-order networks. Thus, suggesting the promise of precision dynamics approach over group-level approaches.

In another recent work, Esfahlani and colleagues also showed bistable brain dynamics at rest using edge-level co-fluctuations. The authors observed the resting brain to oscillate between high- and low-

**Fig. 7 | Replicating results using an independent dataset from the Human Connectome Project (HCP). A** Degree distribution of graphs generated from the real versus null data (from phase randomization and multivariate AR models) revealed heavy (or fat) tail distributions in the real data. **B** Highlight hubs for three representative participants. **C** Annotating Mapper-generated graphs using the relative engagement of a set of canonical large-scale resting state networks (RSNs). Like MSC data, the HCP dataset also revealed that highly connected and central hubs contained brain volumes where no particular RSN was activated, whereas nodes with brain volumes dominating from one particular RSN tend to occupy the peripheral corners of the landscape. Using a variance-based approach, like the MSC-dataset, we again observed a *smooth topographic gradient* in the dynamical

landscape of HCP participants. **D** Traversal in the temporal domain at the single frame level for 30 representative HCP participants. Only showing a sub-set of timeframes for ease of view. Color depicts transitions in brain activation over time frames in terms of dominant individual-specific RSN (or hub-like state). **E** Group averaged distribution of SD values, over all the HCP participants, for hubs (blue) and other nodes (orange) is shown, with SEM as shaded value. **F** Group averaged transition probability matrix derived using Markov chains, indicating the hub-state to be the most sought-after destination from any other RSN-dominated state. Diagonal values were set to zero for ease of visualization. **G** Estimated Markov chain for two representative participants. As evidence the hub-state was observed to be most central and with highest in-degree.

amplitude edge-level co-fluctuations. Further, the authors showed that the relatively short-lived high-amplitude edge co-fluctuations (i) drove the functional organization of resting brain (estimated using functional connectivity; rsFC) (ii) were observed to be highly correlated with high-amplitude BOLD (activity) fluctuations; and (iii) were more similar within than between subjects[78,80]. Although we examined transitions in whole-brain activity (as compared to co-fluctuations between regions), we also observed the amplitude-level dichotomy, such that the peripheral nodes of the landscape contained high-amplitude network-specific activations while the hub-nodes contained mean-level low-amplitude activations. We also found that the co-localization of RSNs (primarily driven peripheral nodes) were highly subject specific.

From the metabolic point of view, Zalesky and colleagues showed that the resting brain dynamically transitions between high- and low-efficiency states[32]. The high efficiency states were characterized by global coordination across brain regions, thus optimizing information processing at a putatively larger expense of metabolic energy. The low efficiency states on the other hand were characterized by lack of global coordination and putatively requiring minimal metabolic expenditure. Although our results are based on whole-brain activation patterns and do not use sliding windows, the whole-brain configurations represented by the hubs could putatively require minimal metabolic expenditure due to the low or close to mean activation amplitude, whereas the configurations represented by the peripheral nodes could potentially require high metabolic expenditure as they show high amplitude network-specific activation. It is important to note that the approaches that focus on co-fluctuations between brain regions might miss brain configurations represented by hub-nodes due to their low-amplitude and putatively low co-fluctuations between brain regions. Future work is required to carefully combine activation-based fluctuations in brain dynamics with fluctuations in coordination across brain regions to better understand how changes in network activations relate to co-fluctuations.

Using a sliding-window based functional connectivity approach, Reinen and colleagues also examined the time-varying organization structure of cortical brain networks at rest[81]. The authors observed a hierarchical structure of network dynamics that was subject-specific and provided evidence for a global attractor state. Interestingly, the functional configuration (i.e., pairwise relations between brain regions) of the global attractor state most closely resembled the average functional configuration of the entire scan. Further, the global attractor state was typified by a relatively flattened profile of within-network connectivity, indicating no particular preference for any RSN. Although our approach is based on whole-brain activation patterns and does not use sliding windows, the whole-brain configurations represented by the hubs (in the Mapper graph) also show muted activation profiles across all RSNs. To further explore the relation between hub states from our work and the global attractor state observed by Reinen et al., we examined the connectivity profile of hub states. We also observed our activation-derived hub states to have muted profiles similar to Reinen et al's global attractor state (Fig. S8).

Future work is required to fully explore the relationship between the time-varying organization of brain activity and connectivity.

Diving deeper, using precision analytics, we revealed rich subject-specific idiosyncrasies. Our approach was developed to examine brain activity dynamics at the single participant level, as opposed to previous approaches that have used group-level data to define states[32,36,77]. Thus, along with precision connectomics data, our precision dynamics approach facilitated finer examination of dynamical organization at rest than done before. For example, although across participants we observed bistable brain dynamics of transitioning between hub and peripheral states, our approach also revealed a large degree of individual variability in terms of the configuration of peripheral nodes. Different combinations of resting state networks dominated peripheral nodes, albeit these combinations were highly subject-specific and consistent across sessions. Further, estimated temporal transition probabilities between RSN-dominated states were also more similar within- than between-participants. Overall, pointing towards future application of our approach in precision medicine.

Examining the traversal on the landscape as well as across the individual timeframes suggest that the brain configurations represented by hub-nodes were putatively acting as a transition state between different parts of the landscape (and respective brain configurations or states). At the single timeframe level, the hub state was also observed to be the most sought-after destination from any other RSN-dominated state. Thus, suggesting a putative intermediary and faciliatory role of the low (or close to mean) amplitude hub states in enabling neural switching between high-amplitude RSN-dominated states. Descriptively, the hub-nodes can be thought of serving a role akin to transportation hubs (e.g., the Grand Central Station for trains), such that these hub-nodes facilitate efficient travel as well as cost-effective transportation architecture. It is also possible that the hub states represent washout (or recovery) configurations of the brain between high-amplitude brain states represented by the peripheral nodes. Future work using our precision dynamics approach in conjunction with theoretical biophysical modeling[82] and neuromodulation experiments[83] is needed to better understand how the hub-states facilitate transitions in the intrinsic brain.

When the Mapper-generated graphs were annotated by variability in mean activation across RSNs, a smooth topographic gradient was consistently observed across all participants. The spontaneous brain activity was observed to be spatiotemporally organized in a continuous gradient with hub- and peripheral-nodes at the opposite ends of the spectrum. Recent work has shown existence of spatial gradients that provide organization principle for anatomical organization of large-scale brain networks as a spectrum from unimodal to heteromodal networks[84]. Here, we provide evidence for a dynamical topographic gradient organizing spontaneous brain activity at rest. Looking forward, our precision dynamics approach can be used to understand differences in temporal organization across various mental health disorders.

In terms of methodological advances, our TDA-based Mapper approach provides a novel avenue to conceptualize fluctuations in brain dynamics at rest, while addressing several limitations with

similarly aimed previous approaches. Broadly speaking, most of the previous approaches conceptualized transitions in the at-rest brain by either estimating inter-regional (or inter-voxel) co-fluctuations over time (e.g., sliding window Pearson's correlation[48], dynamical conditional correlation[49], and multiplication of temporal derivatives[50]) or by exploring brain activations on the basis of sparse events (e.g., co-activation patterns[85], paradigm-free mapping[52] and point process analysis[53]). Further, previous work clustered the observed transitions into a set of configurations (or states) at the group level, thereby putatively missing on the subject-level idiosyncrasies[32,36,77]. Although several key insights were revealed using previous approaches, e.g., bistability of the resting brain[77] and applications in clinical realms have been attempted[86], several methodological limitations were also identified[30,43,87]. First, it is unclear what spatiotemporal scale is ideally suited for studying brain dynamics, i.e., what window length (or threshold for tagging sparse events) is ideal for measuring transitions[30]. Further, a priori knowledge is also required to estimate the number of configurations (or states) during clustering. Second, recent work using linearity preserving surrogate data showed that some of the findings recovered using time-varying analysis could be artifactual due to sampling variability[43,44]. Third, statistical models like HMM also require strict assumptions related to the mutual exclusivity of brain states and require a priori knowledge about number of states[77].

Our Mapper-based approach can work directly at the spatiotemporal scale at which the data were acquired and thus bypasses the issues associated with sliding-window based analysis (e.g., how to choose window-length and reduce artifacts related with sampling variability). Recently, a similar Mapper-based approach was shown to capture and track the task-evoked brain dynamics that matched known ground truth transitions associated with the experimental design[33]. Further, our Mapper-based approach also distinguishes itself from the category of exploring dynamics based on sparse events, because the output does not necessarily assume that brain dynamics arise from only a subset of significant events but permits exploration of the continuous unfolding of dynamics across each time frame. Further, the Mapper-based approach does not require estimation of correlation (or connectivity) between parcellated brain regions and instead use whole-brain activation maps to extract the overall landscape of brain dynamics. Lastly, no assumptions are required to be made regarding mutual exclusivity of brain states or resting state networks. Instead, Mapper generated graphs can be later annotated (e.g., using pie-chart based visualization) to reveal overlapping communities (or states).

Some limitations of our work and associated avenues for future work should also be noted. Although we used a precision individual connectomics dataset to show stable results with ~2.5 hours of resting state fMRI data per individual, realistically, especially in outpatient clinical settings, acquiring that much data from every individual may not be feasible (except perhaps in case of surgical settings). We also replicated the main findings in an independent cohort from the HCP, with ~1 h of rsfMRI data per individual. However, future work is required to examine whether our approach would work with datasets that are not as dense (e.g., traditional rsfMRI scans of 10–20 min of rsfMRI data)—potentially leveraging alternative acquisition paradigms[88]. Another potential limitation and avenue for future work includes combining activation-based dynamics with co-fluctuation of signal across brain regions. New methods are being developed that can provide fluctuations in functional connectivity at the single frame[37], thus in future TDA-based approaches could be used to combine different degrees of interactions between brain regions ranging from brain activations themselves to higher-order interactions. Future work is also required to better understand what purpose the hub state serves in intrinsic dynamics and whether similar hub states can be seen under other states of consciousness (e.g., under anesthesia or sleep). One putative hypothesis could be that the intermittent hub state could

correspond to wash-out state required for the brain before moving from one precise set of brain configuration to the next. Lastly, due to better signal to noise ratio, we restricted our analysis to cortical activity only. Future work is thus required to include sub-cortical structures and cerebellum to better understand their role in the dynamical organization of the brain.

Although the topology of Mapper-generated graphs was largely similar across participants, key subject-specific idiosyncrasies were also observed. For example, which networks (or group of networks) dominated the periphery of the landscape was highly subject-specific and reliable across sessions. Further, the Markov chains, estimated from individual time-frame data, were also observed to be not only subject-specific but also reliable across sessions. These results provide preliminary evidence that our Mapper-related approach contains potential utility for precision medicine approaches. Due to the small number of participants in the MSC dataset and only a moderate group size of the HCP cohort used here, we did not attempt to associate topological properties of Mapper-generated landscapes and trait behavior (e.g., intelligence); as large samples are required for reproducible brain-behavioral phenotypic associations[89]. Future work, using data from large consortia (e.g., leveraging the Adolescent Brain Cognitive Development (ABCD) Study[90]; ($n > 11,000$)) such brain-behavior associations could be examined.

Altogether, we present a novel approach to reveal the rules governing transitions in intrinsic brain activity that could be useful in understanding both typical and atypical cognition. Our work extends previous work both methodologically and conceptually. We observed the dynamical landscape of at-rest brain to contain a shared attractor-like basin that acted like an intermediate state where all canonical resting-state networks were represented equally, while the surrounding periphery had distinct network configurations. Traversal through the landscape suggested continuous evolution of brain activity patterns at rest. Lastly, differences in the landscape architecture were more consistent within than between subjects, providing evidence that this approach contains potential utility for precision medicine approaches.

## Methods
### Datasets

**Midnight scan club (MSC) dataset**. These data were collected from ten healthy, right-handed, young adult subjects (5 females; age: 24–34). One of the subjects is author NUFD, and the remaining subjects were recruited from the Washington University community. Informed consent was obtained from all participants. The study was approved by the Washington University School of Medicine Human Studies Committee and Institutional Review Board. These data were obtained from the OpenNeuro database. Its accession number is ds000224.

For details regarding data acquisition please see Gordon et al. 2017[57]. Briefly, MRI data acquisition for each subject was performed on a Siemens TRIO 3 T scanner over the course of 12 sessions conducted on separate days, each beginning at midnight. Structural MRI was conducted across two separate days. On ten subsequent days, each subject underwent 1.5 h of functional MRI scanning beginning at midnight. In each session, thirty contiguous minutes of resting state fMRI data were acquired, in which subjects visually fixated on a white crosshair presented against a black background. Across all sessions, each subject was scanned for 300 total minutes during the resting state. All functional imaging was performed using a gradient-echo EPI sequence (TR = 2.2 s, TE = 27 ms, flip angle = 90, voxel size = 4 mm × 4 mm × 4 mm, 36 slices).

**Human connectome project (HCP) dataset**. We gathered these data from the Human Connectome Project database[27,91,92] [https://db.humanconnectome.org/]. We specifically chose the $n = 100$ unrelated

cohort (54 females, mean age = 29.1 ± 3.7 years). This cohort of subjects ensures that the participants are not family relatives. As per the HCP protocol guidelines, all participants gave written informed consent for data collection. The HCP scanning protocol was approved by the local Institutional Review Board at Washington University in St. Louis. All experiments were performed in accordance with relevant guidelines and regulations.

A total of 4 resting state fMRI runs were acquired from each participant, where each run was approximately 15 min long. The resting-state fMRI runs (HCP filenames: rfMRI_REST1 and rfMRI_REST2) were acquired in separate sessions on two different days, with two different acquisitions (left to right or LR and right to left or RL) per day[93].

## Preprocessing

**Midnight scan club (MSC).** Preprocessing for these data is described in detail elsewhere[57]. Here, we briefly list the steps. All functional data were preprocessed to reduce artifact and to harmonize data across sessions. All functional data underwent correction for interleaved acquisition, intensity normalization, and head movement. Atlas transformation was computed by registering the mean intensity image from the first BOLD session to Talairach atlas space via the average high-resolution T2-weighted image and average high-resolution T1-weighted image. This atlas transformation, mean field distortion correction, and resampling to 3-mm isotropic atlas space were combined into a single interpolation using FSL's applywarp tool[94].

To reduce spurious variance due to artifacts, further preprocessing was done on each resting state fMRI session. Denoising was accomplished by regression of nuisance time series following a CompCor-like[95] (i.e., component-based) procedure, described in detail elsewhere[96]. Briefly, a design matrix was constructed to include the 6 rigid parameters derived by retrospective motion correction, the global signal averaged over the brain, and orthogonalized waveforms extracted from the ventricles, white matter and extra-cranial tissues (excluding the eyes). Frame censoring (scrubbing) was computed on the basis of both frame-wise displacement (FD) and variance of derivatives (DVARS)[97]. Rigid-body motion parameters were low-pass filtered (<0.1 Hz) prior to FD computation to remove respiratory artifacts in head-motion estimates[98]. The data then were temporally bandpass filtered prior to nuisance regression, retaining frequencies between 0.005 Hz and 0.1 Hz. Censored frames were replaced by linearly interpolated values prior to filtering. The final set of regressors was applied in a single step to the filtered, interpolated BOLD time series. The temporally masked (or censored) frames were then removed for further analysis.

To reveal individual-specific parcellation of the brain, a gradient-based parcellation method was used. See Gordon et al.[57] for more details on this approach. Across all participants, the mean ± SD number of parcels created was 620.8 ± 39.4. The average time course within each resulting parcel was then calculated.

## Human connectome project (HCP)

Minimally processed data were gathered from the HCP database. This minimal processing includes spatial normalization, motion correction, and intensity normalization[99]. We additionally processed these data using fMRIPrep 1.5.9[100].

The fMRIPrep based anatomical preprocessing included correction for intensity non-uniformity (INU) with N4BiasFieldCorrection[101], distributed with ANTs 2.2.0[102], and used as T1w-reference throughout the workflow. The T1w-reference was then skull-stripped with a Nipype implementation of the antsBrainExtraction.sh workflow (from ANTs), using OASIS30ANTs as target template. Brain tissue segmentation of cerebrospinal fluid (CSF), white-matter (WM) and gray-matter (GM) was performed on the brain-extracted T1w using fast (FSL 5.0.9[103]). Volume-based spatial normalization to two standard spaces (MNI152NLin6Asym, MNI152NLin2009cAsym) was performed through nonlinear registration with antsRegistration (ANTs 2.2.0), using brain-extracted versions of both T1w reference and the T1w template.

The fMRIPrep based functional preprocessing included following steps. First, a reference volume and its skull-stripped version were generated using a custom methodology of fMRIPrep. The BOLD reference was then co-registered to the T1w reference using flirt (FSL 5.0.9[94]); with the boundary-based registration cost-function[104]. Co-registration was configured with nine degrees of freedom to account for distortions remaining in the BOLD reference. Head-motion parameters with respect to the BOLD reference (transformation matrices, and six corresponding rotation and translation parameters) are estimated before any spatiotemporal filtering using mcflirt[105] (FSL 5.0.9). The BOLD time-series were resampled onto their original, native space by applying the transforms to correct for head-motion. Several confounding time-series were calculated based on the preprocessed BOLD: framewise displacement (FD), DVARS and three region-wise global signals. FD and DVARS are calculated for each functional run, both using their implementations in Nipype (following the definitions by Power et al.[106]). The three global signals are extracted within the CSF, the WM, and the whole-brain masks. The head-motion estimates calculated in the correction step were also placed within the corresponding confounds file.

Similar to the pre-processing of MSC dataset, here we first calculated temporal masks to flag motion-contaminated frames. We also used a FD > 0.2 mm as threshold to flag a frame as motion contaminated. For each such motion-contaminated frame, we also flagged a back and two forward frames as motion contaminated. Participants were dropped from further analysis, if >20% frames were flagged as motion contaminated. Hence, out of the 100 participants, further analysis was run on n = 76 HCP participants. Following construction of temporal mask for censuring, similar to the MSC data, the HCP data were processed with the following steps: (i) demeaning and detrending, (ii), multiple regression including: whole brain, CSF and white matter signals, and motion regressors derived by Volterra expansion[107], with temporally masked data were ignored during beta estimation, (iii) interpolation across temporally masked frames using linear estimation of the values at censored frames[108] so that continuous data can be passed through (iv) a band-pass filter (0.009 Hz < f < 0.08 Hz). The temporally masked (or censored) frames were then removed for further analysis.

As individual-specific parcellation was not available for the HCP dataset, we used group parcellation from Gordon et al.[76]. The parcellation is based on boundary maps defined using homogeneity of resting state functional connectivity patterns.

## Mapper pipeline

The Mapper pipeline was individually run on each participant. After preprocessing, parcellated time-series (dimension: time-frames x number of parcels) was fed into the Mapper pipeline. These input time-series were concatenated across sessions within participant. For the MSC dataset, the input time-series were concatenated across odd versus even sessions, whereas for the HCP dataset, the input time-series were concatenated across all four available sessions. To harmonize data across sessions, data were z-scored (column-wise) before concatenating across sessions.

Details of Mapper analysis pipeline are presented elsewhere[33,34,66]. Briefly, the Mapper analysis pipeline consists of four main steps. First, Mapper involves embedding the high-dimensional input data into a lower dimension d, using a filter function f. For ease of visualization, we chose d = 2. The choice of filter function dictates what properties of the data are to be preserved in the lower dimensional space. For example, linear filter functions like classical principal component analysis (PCA) could be used to preserve the global variance of the data points in the high dimensional space. However, a large number of

studies using animal models and computational research suggest that inter-regional interactions in the brain are multivariate and nonlinear[82,109,110]. Thus, to better capture the intrinsic geometry of the data, a nonlinear filter function based on neighborhood embedding was used[33]. Thus, instead of measuring Euclidean distances, geodesic (or shortest path) distances were computed between whole-brain configurations (volumes) in the input space. Followed by embedding the graph distances into a d-dimensional Euclidean space, while preserving the intrinsic geometry of the original input. Nonlinear functions like neighborhood embedding allows for preservation of the local structure evident in the original high-dimensional space after projection into a lower dimensional space. Similar functions have been used previously in the field of manifold learning[111–114]. In a recent work, we showed the efficacy of neighborhood embedding in capturing the landscape of whole-brain configurations extracted from a continuous multitask paradigm and task-evoked data from the human connectome project (HCP)[33].

The second step of Mapper performs overlapping n-dimensional binning to allow for compression and reducing the effect of noisy data points. Based on previous work using fMRI data[33], we divided the lower dimensional space into overlapping bins using a resolution parameter (#bins) of 30 for the MSC dataset and 14 for the HCP dataset. The resolution parameter was adjusted based on differences in the temporal resolution of acquisition. The %overlap between bins was kept similar across datasets to 70%. Mapper-generated graphs have been previously shown to be stable for a large variation across parameters for resolution and %overlap[33].

The third step of Mapper includes partial clustering within each bin, where the original high dimensional information is used for coalescing (or separating) data points into nodes in the low-dimensional space. Partial clustering allows to recover the loss of information incurred due to dimensional reduction in step one[34,66]. Lastly, to generate a graphical representation of the "shape" of input data, nodes from different bins are connected if any data points are shared between them.

The Mapper-generated graphs can be annotated (or colored) using meta-information that was not used to construct the graphs. Here, we annotated these graphs using several meta-analytics—ranging from nodal degree to activation in the known large-scale brain networks.

## Topological properties

Several topological properties of the Mapper-generated graphs were studied. We first estimated the nodal degree for each node in the Mapper-generated graphs. In a binary undirected network, the degree, $k_i$, of node $i$ is the number of edges connecting node $i$ with all other $j = 1 \ldots N - 1$ nodes,

$$k_i = \sum_{j \neq i} A_{ij} \qquad (1)$$

The histogram of nodal degrees was then plotted to examine degree distribution derived from real versus null data. In network science, degree distributions can allow us to determine whether the network contains hubs (highly and centrally connected nodes), e.g., fat tail distributions point towards the existence of hub nodes.

Hub nodes in a graph could act as focal points for the convergence and divergence of information in the network. Previous work has suggested that for reliable identification of hubs both degree as well as centrality should be taken into account[73]. Specifically, for degree, we use the cut-off (>21) revealed by comparison of real data with the null data. For centrality, we use the previously prescribed measure of closeness centrality[73]. The closeness centrality of a node is

defined as the inverse of its average shortest path length,

$$C_C(i) = \frac{N - 1}{\sum_{j \neq i} l_{ij}} \qquad (2)$$

where $l_{ij}$ is the shortest path length between nodes $i$ and $j$.

Here, for both the MSC and HCP datasets, we chose nodes with top 1% closeness centrality estimates to define the hub nodes.

## Graph visualization

The Mapper-generated graphs were annotated (or colored) using several features, including topological properties (e.g., nodal degree) or properties derived from the meta-information (e.g., session information). Annotation based on meta-information derived from individual time frames (e.g., session or RSN-based activation) were visualized using a pie-chart based visualization—to present proportional information without averaging data across time frames from each node. A web-based interface was used to interact with the Mapper-generated graphs. This implementation was developed using HTML5, Scalable Vector Graphics (SVG), CSS, and JavaScript. Specifically, we used the D3.js framework (Data-driven documentation; D3) for displaying and annotating individual participants' shape graphs. See our DyNeuSR[34] toolbox for more information.

## Discrete time Markov chains

To better characterize transitions at the single time frame level, we estimated the discrete-time, finite-state, time-homogeneous Markov chains[75] for each participant and data split. Matlab's *dtmc* function was used to estimate these Markov chains, with the empirical count of observed transitions from state $i$ to state $j$ as input. To reduce the effect of head movement related artifact and other artifactual transitions due to stitching even (or odd) sessions together, we ignored transitions associated with frames discarded due to head movement and due to stitching the sessions together.

## Parameter perturbation

Although in the previous work Mapper-generated shape graphs were shown to be robust to a wide-range of parameter perturbation[33], as an additional measure of reliability we again tested the effect of parameter perturbation on the topological properties (e.g., degree distribution) of the Mapper-generated graphs. We varied the two main Mapper parameters—i.e., the number of bins (or resolution, $R$) and percentage of overlap between bins (or gain, $G$)—to generate **121** different variations of the Mapper output for each MSC participant and split of the data. These two binning parameters largely control the overall arrangement of shape graph. Thus, to test whether the topological properties (e.g., degree distribution) is robust in the face of perturbing parameters, we varied $R$ from 25 to 35 (R-5 to R + 5) while $G$ was varied from 65 to 75 (G-5% to G + 5%). Results are shown in the Fig. S6. Overall, the properties were reliably observed in most parameter variations, such that real data was observed to have a fat tail distribution as compared to the null models.

## Null models

To account for linear properties of the data (e.g., serial auto-correlation) and sampling variability issues, we compared Mapper-generated results with two null models, namely, the phase randomized null[68] and the multivariate autoregressive null model[44]. Phase randomization involves randomizing the observed time series by performing Fourier transform, scrambling the phase and then inverting the transform to get the null model. Multivariate autoregressive randomization generates null data by first estimating a single brain parcel x parcel $A_l$ matrix, for each lag $l$. Here, an AR order of $p = 1$ was used, as prescribed by earlier work[44]. The autocorrelation function, power spectrum, and other linear properties are preserved under both phase randomization

and multivariate autoregressive randomization. Several instances of null data were generated for each participant separately (25 per participant and per split of the data). We used previously published Matlab-based scripts to generate both phase randomization and multivariate autoregressive null model simulations[44]. These scripts are available to download from the Github repository (https://github.com/ThomasYeoLab/CBIG/blob/master/stable_projects/fMRI_dynamics/Liegeois2017_Surrogates/).

## Reporting summary

Further information on research design is available in the Nature Research Reporting Summary linked to this article.

## Data availability

The MSC data used in this work were originally collected by Gordon et al.[57] and is available for download at https://openneuro.org/datasets/ds000224/versions/1.0.3. The second dataset was originally collected as part of the Human Connectome Project (HCP)[115]. We gathered these data directly from the HCP website (https://db.humanconnectome.org). Source data for figures are provided with this paper. Source data are provided with this paper.

## Code availability

The code required for generating the Mapper graphs and corresponding figures presented in the paper is made available at https://github.com/braindynamicslab/tda-msc-rsfMRI. The Zenodo doi for this code is https://zenodo.org/badge/latestdoi/285977733. For graph theoretical analysis, we used the Brain Connectivity Toolbox (https://sites.google.com/site/bctnet/). For analyzing fMRI data, we used FSL toolbox v6.0 (available here https://fsl.fmrib.ox.ac.uk/fsl/fslwiki). Brain activation overlays were created using the Connectome Workbench Viewer (wb_view; https://www.humanconnectome.org/software/connectome-workbench).

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

## Acknowledgements

This work was supported by an NIH Director's New Innovator Award (DP2; MH119735), an NIH Career Development Award (K99/R00; MH104605), and an MCHRI Faculty Scholar Award to M.S. The Midnight Scan Club data acquisition was supported by grants from the National Institute of Health (NS088590); the Jacobs Foundation (2016121703); and the Kiwanis Neuroscience Research Foundation to N.U.F.D. Funding for the Human Connectome Project data acquisition were provided by the 16 NIH Institutes and Centers that support the NIH Blueprint for Neuroscience Research (as part of the Human Connectome Project, WU-Minn Consortium; Principal Investigators: David Van Essen and Kamil Ugurbil; 1U54MH091657) and by the McDonnell Center for Systems Neuroscience at Washington University. R.L. was supported by the National Centre of Competence in Research - Evolving Language grant (51NF40_180888). D.F. was supported by grants from the National Institute of Health (MH096773, MH115357, and DA041148).

## Author contributions

M.S.: conceptualization, methodology, software, validation, investigation, visualization, and writing. J.M.S.: methodology, validation, interpretation, and writing. R.L.: methodology, validation, interpretation, and writing. N.U.F.D: methodology, validation, data curation for the MSC data, interpretation, and writing. D.F.: conceptualization, methodology, validation, interpretation, writing, and supervision.

## Competing interests

The authors declare no competing interests.
