## [Peer Review File · Nature Communications]

REVIEWER COMMENTS

Reviewer #1 (Remarks to the Author):

In the present manuscript the authors present a topological data analysis-based mapping approach, using it to examine principles of participant-level functional dynamics at rest. In doing so they report the presence of a topographic hub, where all resting-state networks are uniformly activated, that potentially acts as transition state that links alternate network configurations.

I commend the authors on their interesting approach incorporates linear and nonlinear spatiotemporal similarities of whole brain dynamics, which bypasses the limitations of traditional methods based on inter-regional co-activation. However, I have some concerns regarding the methods and associated conclusions which I detail below. In particular, I am not fully convinced that the hub-state in the topographic landscape necessarily serves as transitional state in real time, and that the continuous gradient in the topographic space suggest that the transitions between brain states are also continuous in time domain.

In result 2.2, high nodal degree and closeness centrality are evidence that the hub state are more frequently visited – it is possible that this configuration serves as switch between states, but it is also possible that this state appears more frequently than all others. To conclude on its switching role, further evidence is needed to exclude the alternative possibility.

The use of the term “hub nodes” to refer to whole-brain configurations may be confusing to some readers. Hub nodes typically reflect to specific parcels in the literature rather than a connectome-level connectivity pattern.

In figure 2 B the MSC-01 odd and even sessions seem to display visually distinct patterns. Is this expected? Or should results be more reliable at the subject-level?

It is encouraging to see the stable continuous topographical gradient across subjects and the replication sample, and it is interesting in Fig 5. that paths from different peripheral node to hub nodes reflect continuous decay of the dominant resting-state network in the peripheral node. However, the way that nodes are mapped back to time domain lost this continuity. It seems to me that it is more like a dichotomization of the hub nodes and peripheral nodes in time domain. “For nodes dominated by any particular RSN, the encompassing TRs were assigned the dominant RSN” keeps the peripheral nodes feature, where the dominant RSN reaches peak in its activation relative to the activations of all other

RSNs; whereas “For hub nodes, where RSNs were uniformly distributed, the encompassing TRs were assigned a new label (Hub)” extract the tail part of the RSN dominance.

This is to say, the current evidence in section 2.5 support that the hub states are the most frequently visited after other states in time domain (in most but perhaps not all of the MSC sample). To make inference on its transitional and continuous nature, I would suggest additional plots focusing on the single dominant RSNs and calculate its activation relative to mean activations of all other RSNs, then propagate this ratio back to the time domain, see if the evolution between peak and tail of this RSN dominance is continuous. Then the authors could overlay the hub node onto this plot to check if the hubs match up with the tails of the RSN dominance.

While the authors do a good job of summarizing the literature, some key work in this domain is neglected. In particular, although the approach is distinct (current mapping approach vs sliding window), many of the present results echo the work of Reinen et al. 2018 also published in Nat Comms. There, the authors demonstrated the presence of hierarchical structure to network dynamics at rest, provide evidence for an attractor state, and show that network dynamics are more consistent within than between subjects. The authors should interpret their current work in light of these prior discoveries.

Relatedly, can the authors comment on the relationship between their “hub nodes” and the typical network profile observed in a resting-state analysis? Do these nodes reflect a closer approximation of the “average state” than the non-hub profiles. Similar to the “attractor state” of Reinen et al.

Another concern is in result 2.6, Fig. 7E. It is unclear how the 30 subjects are selected; it would be better to explain the subjects are randomly selected if this is the case.

Minor point, but it is unclear to me what “highest” means in this context, “Mapper has been previously shown to capture task-evoked transitions in the whole-brain activity patterns at the highest spatiotemporal resolution” (Page 4).

On page 5 the authors suggest “Second, overlapping d -dimensional binning is performed to allow for compression and to reduce any destructive effects of noise.” Are they certain “any” impact of noise has been fully removed from their data? As this would be quite difficult to prove, they should like reword this sentence.

Reviewer #2 (Remarks to the Author):

The authors aim at revealing the overall landscape of at-rest whole-brain configurations(or states) at the single individual level using a precision dynamics approach and the Midnight Scan Club(MSC). This work hopes to interpret the rules that govern transitions in brain activity at rest and used the TDA-based Mapper approach. Although the figures of dynamic mapper are fancy and beautiful, some problems still need to clarify.

1. Why do the odd sessions and even sessions look so different from each other for the same subjects in Figure 3, but their correlation or some measurement is high?
2. Actually, the individual-level reproducibility is still measured by regional similarity, similar to FC fingerprints, but what are the additional advantages beyond the moving landscape organization for the dynamic landscape?
3. How about the time efficiency of the TDA? Can we use in a relatively short time course instead of 5 hours? And need at least how short the results will be stable? In most cases, people will not have that long time course.
4. Finally, except for the temporal transition of different hub arrays, is it possible to link the landscape organization with real brain anatomical mapping? Which may make the results more interpretable?

Dear Reviewers,

Thank you for reviewing our manuscript “*Precision dynamical mapping using topological data analysis reveals a unique hub-like transition state at rest*”. We appreciate your enthusiasm for the question addressed in this study and for our approach. In response to your constructive comments, we have made several changes to this manuscript. We hope you will find the revisions to this manuscript satisfactory.

Please also note that while performing additional analysis, as recommended by you, we discovered an **error** in our code. This error was related to reading neuroimaging data (i.e., cifti formatted values), we have thus fixed this error and re-ran all the analyses from scratch. Thankfully, all the previous results were not only **replicated** but we also have **improved** results. However, all the figures are re-generated and hence might look slightly different from the last version. We apologize for the error and any related inconvenience. Specifically, the error was narrowed down to a third-party code for reading cifti data (function name: `ft_read_cifti_mod`). We have now replaced that function with standard `cifti_read` function from the Human Connectome Project GitHub site (more information here <https://github.com/Washington-University>).

The response letter is organized as follows: we present our response after each comment. We have also highlighted the edited text in the revised manuscript (with **blue color**) for ease of review.

Sincerely,

Manish Saggarr

Reviewer #1 (Remarks to the Author):

General remark: In the present manuscript, the authors present a topological data analysis-based mapping approach, using it to examine principles of participant-level functional dynamics at rest. In doing so they report the presence of a topographic hub, where all resting-state networks are uniformly activated, that potentially acts as a transition state that links alternate network configurations.

Comment 1: I commend the authors on their interesting approach incorporates linear and nonlinear spatiotemporal similarities of whole-brain dynamics, which bypasses the limitations of traditional methods based on inter-regional co-activation. However, I have some concerns regarding the methods and associated conclusions which I detail below. In particular, I am not fully convinced that the hub-state in the topographic landscape necessarily serves as a transitional state in real-time and that the continuous gradient in the topographic space suggests that the transitions between brain states are also continuous in the time domain.

Response: We thank the reviewer for the encouragement. We have now added new plots/analyses to better depict the continual/transitional nature of hub-states (lines: 436-448 in the revised manuscript and a new supplementary figure – **Fig. S7**). Please also see our more detailed response to this concern in Comment #5, below. Thanks to the suggested new analysis by the reviewer, we indeed find further evidence for the continual nature of the transition between brain states via the hub-states.

Comment 2: In result 2.2, high nodal degree and closeness centrality are evidence that the hub state are more frequently visited – it is possible that this configuration serves as switch between states, but it is also possible that this state appears more frequently than all others. To conclude on its switching role, further evidence is needed to exclude the alternative possibility.

Response: We agree with the reviewer, please see our detailed response to this concern in Comment #5. Briefly, as suggested by the reviewer, we added new analyses/plots to examine the transitional and continuous interplay between hub states and RSN-dominated states. While focusing on one dominating RSN at a time, we first annotated nodes in the Mapper graph based on dominant RSN's activation relative to other RSNs. Followed by propagating this Mapper node-based mean value to the corresponding individual timeframes (TRs), to examine whether the hub-states appear at the tails of RSN-dominance or are simply more frequent. We found

evidence for the switching role of hub-states, as they tend to appear more likely at the tails of RSN-dominance (see **Fig. S7** pasted in response to Comment #5).

Comment 3: The use of the term “hub nodes” to refer to whole-brain configurations may be confusing to some readers. Hub nodes typically reflect to specific parcels in the literature rather than a connectome-level connectivity pattern.

Response: We agree with the reviewer. To avoid unnecessary confusion, we have replaced the term “hub nodes” with the term “hubs” (when referring to nodes in a graph) or “hub states” (when referring to transition states of the brain), wherever possible in the revised manuscript.

Comment 4: In figure 2 B the MSC-01 odd and even sessions seem to display visually distinct patterns. Is this expected? Or should results be more reliable at the subject level?

Response: Our TDA-based Mapper approach generates a representation of the underlying shape (or manifold) of the data. However, even with long rsfMRI runs, we cannot expect to capture all possible whole-brain configurations of brain activity. Thus, only a sample of possible whole-brain configurations is captured within each scan session. Further, the nature of resting-state experiments (i.e., no task constraints) implies the presence of larger variation in whole-brain configurations (Shehzad et al. 2009, *Cerebral Cortex*). For these reasons, we did not expect the shape graphs to be visually identical across sessions. Instead, we expected the topological (e.g., degree distribution), topographical (e.g., which RSNs dominate periphery, or which RSNs co-activate), and temporal properties (e.g., transition probabilities) to replicate at the single individual level. In the original version of the manuscript, we reported the replication of topographical (**Fig. 4C**) as well as temporal properties (**Fig. 6E**). We have also added replication of topological properties (i.e., degree distribution) across sessions within each individual, that is, we found degree distributions to be more similar across sessions (odd vs even) within participants as compared to between participants ($F(1,18)=5.31$, $p=0.034$). We have amended the text to better clarify these results (pasted below for ease).

Lines 343-349: In addition to RSN-based topography, subject specificity was also observed in terms of the topological properties of the Mapper-generated landscapes. For example, the degree distribution of Mapper-generated graphs was more similar between sessions within a participant than across participants ($F(1,18)=5.31$, $p=0.034$). Also, the proportion of hubs was similar across splits of the data (i.e., odd vs. even sessions; $F(1,18)=1.73$, $p=0.2$). Thus, suggesting, both topographical and topological properties of the Mapper-generated landscapes were subject-specific and stable across sessions.

Comment 5: It is encouraging to see the stable continuous topographical gradient across subjects and the replication sample, and it is interesting in Fig 5. that paths from different peripheral nodes to hub nodes reflect continuous decay of the dominant resting-state network in the peripheral node. However, the way that nodes are mapped back to the time domain lost this continuity. It seems to me that it is more like a dichotomization of the hub nodes and peripheral nodes in the time domain. “For nodes dominated by any particular RSN, the encompassing TRs were assigned the dominant RSN” keeps the peripheral nodes feature, where the dominant RSN reaches a peak in its activation relative to the activations of all other RSNs; whereas “For hub nodes, where RSNs were uniformly distributed, the encompassing TRs were assigned a new label (Hub)” extract the tail part of the RSN dominance. This is to say, the current evidence in section 2.5 supports that the hub states are the most frequently visited after other states in the time domain (in most but perhaps not all of the MSC samples). To make inference on its *transitional and continuous nature*, I would suggest additional plots focusing on the single dominant RSNs and calculating its activation relative to mean activations of all other RSNs, then propagate this ratio back to the time domain, see if the evolution between peak and tail of this RSN dominance is continuous. Then the authors could overlay the hub node onto this plot to check if the hubs match up with the tails of the RSN dominance.

Response: We thank the reviewer for this suggestion. As suggested by the reviewer, we added new analyses/plots to examine the transitional and continuous interplay between hub states and RSN-dominated states. While focusing on one dominant RSN at a time, we first estimated its activation relative to mean activations of all other RSNs and annotated the nodes in the Mapper graph accordingly (see the new **Fig. S7A**, pasted below for ease). As expected, for each dominant RSN, we observed a gradient of mean activation across nodes in the Mapper graph – such that peripheral nodes contained timeframes (or TRs) with the highest activation, while more central nodes contained TRs with low activation. To evaluate the interplay between hub-states and dominant RSN in the time domain, we propagated the nodal mean from the Mapper nodes to the corresponding TRs. Important to note, as suggested by the reviewer, instead of propagating the information dichotomously (i.e., labeling every TR with dominating network vs. hub state), we propagated mean activation values of dominating RSN to the timeframes. We later overlaid hub-state TRs on the x-axis to denote which TRs belong to the hub states. As evident in the **Fig. S7B**, activation of each of the three dominating networks (default mode,

frontoparietal, and cingulo-opercular) are continuous in nature and importantly, the hub-states are only present at the tails of RSN dominance. To quantify this inverse relation between RSN dominance and hub-states, we estimated the temporal correlation between RSN mean amplitude and hub-state occurrences. As expected, predominantly negative relations were observed between the two for all participants and across sessions, suggesting that the hub-states tend to appear in the tails of RSN dominance and putatively trigger transitions between RSNs (see **Fig. S7D** for a histogram of correlation values across the 10 MSC participants). These new analyses are added to the revised manuscript (pasted below for ease) and supplementary figures (**Fig. S7**).

Lines 436-448: To further confirm the *transitional and continuous* interplay between hub states and RSN-dominated states, we examined whether the hub states appear at the tail ends of RSN-dominance in the time domain (i.e., at the level of individual brain volumes). For this analysis, instead of propagating the RSN-dominance vs. hub state dichotomously into the time domain (i.e., labeling every TR with dominating network or a hub state), we propagated mean activation values of dominating RSN Mapper nodes to the timeframes. The continuous evolution of RSN dominance was observed at the timeframe level and hub states were found more likely to be present at the tails of RSN dominance – providing further evidence for the transitory nature of hub states (**Fig. S7A-C**). To quantify this inverse relation between RSN dominance and hub states, we estimated temporal correlation between RSN mean amplitude and hub state occurrences across participants. Predominantly negative relations were observed between the two for all participants and across sessions (**Fig. S7D**), suggesting that the hub states tend to appear in the tails of RSN dominance and putatively trigger transitions between RSNs.

Fig. S7: The nodal mean of each dominant network was propagated into time domain (individual TRs) to examine the continuous and transitory nature of RSN-dominance vs hub-states. **[A]** Mapper nodes are annotated by activation in three dominant RSNs relative to mean activations of all other RSNs for one representative participant (MSC-01, odd sessions). As expected, for each dominant RSN, we observed a gradient of mean activation across nodes in the Mapper graph – such that peripheral nodes contained timeframes (or TRs) with the highest activation, while more central nodes contained TRs with low activation. As evident in **[B]**, the activation of each of the three dominating networks (default mode, frontoparietal, and cingulo-opercular) are continuous in nature and, importantly, the hub-states tend to appear at the tails of RSN dominance – putatively triggering transitions between RSNs. **[C]** Cortical activations are shown for three representative TRs for each dominant network. **[D]** Histogram of temporal correlation between the mean amplitude of dominant RSNs and hub-state occurrences across all ten participants (separately shown for odd and even sessions). As evident, negative relation between the occurrence of hub-states and activation in one or more RSNs was observed.

Comment 6: While the authors do a good job of summarizing the literature, some key work in this domain is neglected. In particular, although the approach is distinct (current mapping

approach vs sliding window), many of the present results echo the work of Reinen et al. 2018 also published in Nat Comms. There, the authors demonstrated the presence of hierarchical structure to network dynamics at rest, provide evidence for an attractor state, and show that network dynamics are more consistent within than between subjects. The authors should interpret their current work in light of these prior discoveries.

Relatedly, can the authors comment on the relationship between their “hub nodes” and the typical network profile observed in a resting-state analysis? Do these nodes reflect a closer approximation of the “average state” than the non-hub profiles? Similar to the “attractor state” of Reinen et al.

Response: We thank the reviewer for this reference. We have now added and compared Reinen et al findings with ours. We also examined the functional connectivity profile for hub states. Interestingly, we observed our activation-derived hub states to have muted (i.e., uniform across all RSNs) within network profile, similar to Reinen et al’s global attractor state. Although future work is required to fully explore the relationship between the time-varying organization of brain activity and connectivity, we have added these new results and interpretations to the revised manuscript (lines: 642-656; **Fig. S8**; pasted below for ease).

Lines 642-656: Using a sliding-window-based functional connectivity approach, Reinen and colleagues also examined the time-varying organization structure of cortical brain networks at rest (Reinen *et al.*, 2018). The authors observed a hierarchical structure of network dynamics that was subject-specific and provided evidence for a global attractor state. Interestingly, the functional configuration (i.e., pairwise relations between brain regions) of the global attractor state most closely resembled the average functional configuration of the entire scan. Further, the global attractor state was typified by a relatively flattened profile of within-network connectivity, indicating no particular preference for any RSN. Although our approach is based on whole-brain activation patterns and does not use sliding windows, the whole-brain configurations represented by the hubs (in the Mapper graph) also show muted activation profiles across all RSNs. To further explore the relation between hub states from our work and the global attractor state observed by Reinen et al, we examined the connectivity profile of hub states. We also observed our activation-derived hub states to have muted profiles similar to Reinen et al’s global attractor state (**Fig. S8**). Future work is required to fully explore the relationship between the time-varying organization of brain activity and connectivity.

Fig. S8: Examining connectivity profile of hub states. [A] Functional connectivity derived from timeframes of hub Mapper nodes for one representative participant (MSC01, odd sessions). The connectivity matrix is organized by RSNs. Uniform within network connectivity is observed across all networks. [B] Spider charts showing within-network connectivity (derived from hubs) across all 10 participants, separately for odd and even sessions. Although Mapper graphs were generated using activation data (and not connectivity estimates), within network functional connectivity derived from hubs also suggest no preference for any RSN.

Comment 7: Another concern is in result 2.6, Fig. 7E. It is unclear how the 30 subjects are selected; it would be better to explain the subjects are randomly selected if this is the case.

Response: We apologize for the lack of clarity. We simply took the first 30 subjects. We have added this clarification to the text (lines 518-519; pasted below).

Lines 518-519: Fig. 7E depicts RSN-based labels for each TR, across the 30 representative HCP participants (randomly first 30 were chosen).

Comment 8: Minor point, but it is unclear to me what “highest” means in this context, “Mapper has been previously shown to capture task-evoked transitions in the whole-brain activity patterns at the highest spatiotemporal resolution” (Page 4).

Response: By highest we simply meant the highest available resolution at which the data were acquired. We have modified this phrase to clarify the same (lines: 138-141; pasted below).

Lines 138-141: Mapper has been previously shown to capture task-evoked transitions in the whole-brain activity patterns at the highest available spatiotemporal resolution, limited only by acquisition parameters (Saggar *et al.*, 2018).

Comment 9: On page 5 the authors suggest “Second, overlapping d -dimensional binning is performed to allow for compression and to reduce any destructive effects of noise.” Are they certain “any” impact of noise has been fully removed from their data? As this would be quite difficult to prove, they should like to reword this sentence.

Response: We thank the reviewer for pointing this out. We have now reworded the sentence as follows: “*Second, overlapping d -dimensional binning is performed to allow for compression and to putatively increase reliability (by reducing noise-related perturbations).*” (lines: 168-169).

Reviewer #2 (Remarks to the Author):

General remark: The authors aim at revealing the overall landscape of at-rest whole-brain configurations (or states) at the single individual level using a precision dynamics approach and the Midnight Scan Club (MSC). This work hopes to interpret the rules that govern transitions in brain activity at rest and used the TDA-based Mapper approach. Although the figures of the dynamic mapper are fancy and beautiful, some problems still need to clarify.

Comment 1: Why do the odd sessions and even sessions look so different from each other for the same subjects in Figure 3, but their correlation or some measurement is high?

Response: Our TDA-based Mapper approach generates a representation of the underlying shape (or manifold) of the data. However, even with long rsfMRI runs, we can't expect to capture all possible whole-brain configurations of brain activity. Thus, only a sample of possible whole-brain configurations is captured within each scan session. Further, the nature of resting-state experiments (i.e., no task constraints) implies the presence of larger variation in whole-brain configurations (Shehzad et al. 2009, *Cerebral Cortex*). For these reasons, we did not expect the shape graphs to be visually identical across sessions. Instead, we expected the topological (e.g., degree distribution), topographical (e.g., which RSNs dominate periphery, or which RSNs co-activate), and temporal properties (e.g., transition probabilities) to replicate at the single individual level. In the original version of the manuscript, we reported the replication of topographical (**Fig. 4C**) as well as temporal properties (**Fig. 6E**). We have also added replication of topological properties (i.e., degree distribution) across sessions within each individual, that is, we found degree distributions to be more similar across sessions (odd vs even) within participants as compared to between participants ($F(1,18)=5.31$, $p=0.034$). We have amended the text to better clarify these results (pasted below for ease).

Lines 343-349: In addition to RSN-based topography, subject specificity was also observed in terms of the topological properties of the Mapper-generated landscapes. For example, the degree distribution of Mapper-generated graphs was more similar between sessions within a participant than across participants ($F(1,18)=5.31$, $p=0.034$). Also, the proportion of hubs was similar across splits of the data (i.e., odd vs. even sessions; $F(1,18)=1.73$, $p=0.2$). Thus, suggesting, both topographical and topological properties of the Mapper-generated landscapes were subject-specific and stable across sessions.

Comment 2: Actually, the individual-level reproducibility is still measured by regional similarity, similar to FC fingerprints, but what are the additional advantages beyond the moving landscape organization for the dynamic landscape?

Response: The individual-level reproducibility is not directly measured by regional similarity (i.e., co-fluctuations between brain regions) as done typically in case of FC fingerprinting approaches, but by co-presence/activation of networks on the manifold. Further, our approach shows within-participant similarity across topological (e.g., degree distribution), topographical (e.g., which RSNs dominate periphery, or which RSNs co-activate), and temporal properties (e.g., transition probabilities) to replicate at the single individual level. In the original version of the manuscript, we reported the replication of topographical (**Fig. 4C**) as well as temporal properties (**Fig. 6E**). We have also added similarity across topological properties (i.e., degree distribution, number of hub-nodes, etc.) across sessions within-individual.

Comment 3: How about the time efficiency of the TDA? Can we use in a relatively short time course instead of 5 hours? And need at least how short the results will be stable? In most cases, people will not have that long time course.

Response: We thank the reviewer for raising the question of the temporal efficiency of this TDA representation, which is indeed important. We first note that the main goal of our study was to unveil the brain's intrinsic organization using the best possible dataset available (hence the use of the full MSC dataset). This being said, our results were replicated in two datasets with fewer time points: the MSC half split (~2.5 hours of rsfMRI) and the independent HCP dataset (~48 min of rsfMRI). HCP-style data acquisition is getting more common these days which shows that our approach can also be leveraged in such typical acquisition settings. The impact of further decreasing the number of time points raises other challenges and we now elaborate on this in the discussion section (lines: 734-740; pasted below for ease).

Lines 734-740: Although we used a precision individual connectomics dataset to show stable results with ~2.5 hours of resting state fMRI data per individual, realistically, especially in outpatient clinical settings, acquiring that much data from every individual may not be feasible (except perhaps in case of surgical settings). We also replicated the main findings in an

independent cohort from the HCP, with ~1 hour of rsfMRI data per individual. However, future work is required to examine whether our approach would work with datasets that are not as dense (e.g., traditional rsfMRI scans of 10-20 min of rsfMRI data) – potentially leveraging alternative acquisition paradigms (Lynch *et al.*, 2020).

Comment 4: Finally, except for the temporal transition of different hub arrays, is it possible to link the landscape organization with real brain anatomical mapping? Which may make the results more interpretable?

Response: We agree with the reviewer that anchoring our results into the anatomical organization would increase interpretability. In the revised version, we provide such anchoring at several places in the results section. For example, in the revised Fig. 3D, we now provide improved cortical activation maps for peripheral nodes, with clear borders for individual specific RSNs. Further, in the revised **Fig. 5** (pasted below for ease), we show whole-brain activity (on cortical surfaces) for several nodes from one of the manifold trajectories. Due to this added anchoring on cortical surface, it is easier to see how peripheral nodes are dominated by activation in the default mode network and traversal towards the hub nodes result in reduced RSN activity.

Similarly, in the new supplementary **Fig. S7** (pasted below for ease), we show anatomical mapping at the level of single timeframes to show how Mapper can help tease apart the continual nature of RSN dominance and transitions into hub states.

Finally, in order to provide further interpretability of the nature of hub states, we also computed the corresponding functional connectivity profile (see supplementary **Fig. S8**, pasted here under for ease).

Fig. 5: Annotating the traversal on Mapper-generated landscape using a variance-based approach revealed a dynamical *topographic gradient*. [A] Depicting traversal on the Mapper-generated landscape from peripheral (RSN-dominated) nodes towards centrally located hubs. Three putative trajectories are highlighted on the Mapper graph, corresponding to three dominating RSNs for MSC-01 participant. For all three trajectories, activation across RSNs (as box plots) and mean whole-brain activity (on cortical surfaces) is shown for multiple nodes. As evident, peripheral nodes are dominated by activation in one of the RSNs and traversal towards the hubs result in reduced RSN activity. [B] Annotating Mapper-generated graphs using variance-based approach, i.e., coloring nodes based on the amount of variance (or S.D.) across

mean RSNs activation, revealed a dynamical *topographic* gradient. Here, we show variance-based annotation of Mapper graphs for four participants from the MSC dataset (odd sessions). The topographic gradient was observed consistently across participants and for both even and odd sessions (see Fig. S4). [C] Group averaged distribution of S.D. values, over ten MSC participants, for hubs (blue) and other nodes (orange) is shown, with S.E.M. as shaded value. Evidently, the hubs had significantly low variance across mean RSN activation (indicating uniformly distributed RSN), while the non-hub nodes were highly variant across mean RSN activation.

Fig. S7: The nodal mean of each dominant network was propagated into time domain (individual TRs) to examine the continuous and transitory nature of RSN-dominance vs hub-states. [A] Mapper nodes are annotated by activation in three dominant RSNs relative to mean activations of all other RSNs for one representative participant (MSC-01, odd sessions). As expected, for each dominant RSN, we observed a gradient of mean activation across nodes in the Mapper graph – such that peripheral nodes contained timeframes (or TRs) with the highest activation, while more central nodes contained TRs with low activation. As evident in [B], the activation of each of the three dominating networks (default mode, frontoparietal, and cingulo-opercular) are continuous in nature and, importantly, the hub-states tend to appear at the tails of RSN dominance – putatively triggering transitions between RSNs. [C] Cortical activations are shown for three representative TRs for each dominant network. [D] Histogram of

temporal correlation between the mean amplitude of dominant RSNs and hub-state occurrences across all ten participants (separately shown for odd and even sessions). As evident, negative relation between the occurrence of hub-states and activation in one or more RSNs was observed.

Fig. S8: Examining connectivity profile of hub states. [A] Functional connectivity derived from timeframes of hub Mapper nodes for one representative participant (MSC01, odd sessions). The connectivity matrix is organized by RSNs. Uniform within network connectivity is observed across all networks. [B] Spider charts showing within-network connectivity (derived from hubs) across all 10 participants, separately for odd and even sessions. Although Mapper graphs were generated using activation data (and not connectivity estimates), within network functional connectivity derived from hubs also suggest no preference for any RSN.

REVIEWERS' COMMENTS

Reviewer #1 (Remarks to the Author):

I served as a reviewer on the prior submission. The authors have been quiet responsive to my previous comments and I commend them on the excellent set of discoveries. I look forward to seeing this manuscript in print. My remaining, more minor, thoughts and concerns are detailed below.

The mean activation propagated to TRs, and the mostly negative correlation between the mean amplitude of dominant RSNs and hub-state occurrences seems convincing as continuous. In Fig. S7B, I assume the larger circles in the x-axis are HUB, but they are colored as white, which is inconsistent with the legend in gray.

Reviewer #2 (Remarks to the Author):

The authors have addressed most of my concerns and made great improvement on the interpretability of the method, for example, Fig. 5 provides the vivid linkage of the traversal on Mapper-generated landscape with specific brain networks. The only drawback is that the abstract and figure legends still do not link with brain networks or neuroimaging findings direct enough, which is difficult for unprofessional readers to understand. The authors should make more effort on this part to make the findings more interpretable for brain imaging readers. Furthermore, the authors are suggested to cite more findings of dynamic functional networks in the introduction. Such as

Z Fu, A Iraj, JA Turner, J Sui, R Miller, GD Pearlson, VD Calhoun. Dynamic state with covarying brain activity-connectivity: On the pathophysiology of schizophrenia. *Neuroimage* 224, 117385

J Sui, R Jiang, J Bustillo, V Calhoun. Neuroimaging-based individualized prediction of cognition and behavior for mental disorders and health: methods and promises. *2020.Biological psychiatry* 88 (11), 818-828

Dear Reviewers,

Thank you for reviewing our revised manuscript. We appreciate your second review and comments. In response to your constructive comments, we have made the requested changes to this manuscript. We hope you will find the revisions to this manuscript satisfactory.

The response letter is organized as follows: we present our response after each comment. We have also highlighted the edited text in the revised manuscript (with blue color) for ease of review.

Sincerely,

Manish Saggar

Reviewer #1 (Remarks to the Author):

General remark: I served as a reviewer on the prior submission. The authors have been quite responsive to my previous comments, and I commend them on the excellent set of discoveries. I look forward to seeing this manuscript in print. My remaining, more minor, thoughts and concerns are detailed below.

Response: We thank the reviewer for the encouragement.

Comment 1: The mean activation propagated to TRs, and the mostly negative correlation between the mean amplitude of dominant RSNs and hub-state occurrences seems convincing as continuous. In Fig. S7B, I assume the larger circles in the x-axis are HUB, but they are colored as white, which is inconsistent with the legend in gray.

Response: We thank the reviewer for pointing out the inconsistency. We have now revised the Fig.S7B legend to state the hubs are colored as white (and not gray).

Reviewer #2 (Remarks to the Author):

General remark: The authors have addressed most of my concerns and made great improvement on the interpretability of the method, for example, Fig. 5 provides the vivid linkage of the traversal on Mapper-generated landscape with specific brain networks.

Response: We thank the reviewer for the encouragement.

Comment 1: The only drawback is that the abstract and figure legends still do not link with brain networks or neuroimaging findings direct enough, which is difficult for unprofessional readers to understand. The authors should make more effort on this part to make the findings more interpretable for brain imaging readers.

Response: We thank the reviewer for pointing this out. Accordingly, we have now revised the abstract (pasted below for ease). We would also like to humbly point out to the reviewer that we did our best to interpret our findings in relation with previous neuroimaging studies and dedicated a large portion of the Discussion section (lines: 435-589) on it.

Revised abstract: In the absence of external stimuli, neural activity continuously evolves from one configuration to another. Whether these transitions or explorations follow some underlying arrangement or lack a predictable ordered plan remains to be determined. Here, using fMRI data from highly sampled individuals (~5 hours of resting-state data per individual), we aimed to reveal the rules that govern transitions in brain activity at rest. Our Topological Data Analysis based Mapper approach characterized a highly visited transition state of the brain that acts as a switch between different neural configurations to organize the spontaneous brain activity. Further, while the transition state was characterized by a uniform representation of canonical resting-state networks (RSNs), the periphery of the landscape was dominated by a subject-specific combination of RSNs. Altogether, we revealed rules or principles that organize spontaneous brain activity using a precision dynamics approach.

Comment 2: Furthermore, the authors are suggested to cite more findings of dynamic functional networks in the introduction. Such as

1. Z Fu, A Iraj, JA Turner, J Sui, R Miller, GD Pearlson, VD Calhoun. Dynamic state with covarying brain activity-connectivity: On the pathophysiology of schizophrenia. *Neuroimage* 224, 117385
2. J Sui, R Jiang, J Bustillo, V Calhoun. Neuroimaging-based individualized prediction of cognition and behavior for mental disorders and health: methods and promises. *2020.Biological psychiatry* 88 (11), 818-828

Response: We thank the reviewer for pointing out these references. We have now added them in the Introduction section.